# Convergence Guarantees for RMSProp and ADAM in Non-Convex Optimization and an Empirical Comparison to Nesterov Acceleration

## Abstract

RMSProp and ADAM continue to be extremely popular algorithms for training neural nets but their theoretical convergence properties have remained unclear. Further, recent work has seemed to suggest that these algorithms have worse generalization properties when compared to carefully tuned stochastic gradient descent or its momentum variants. In this work, we make progress towards a deeper understanding of ADAM and RMSProp in two ways. First, we provide proofs that these adaptive gradient algorithms are guaranteed to reach criticality for smooth non-convex objectives, and we give bounds on the running time.

Next we design experiments to empirically study the convergence and generalization properties of RMSProp and ADAM against Nesterov's Accelerated Gradient method on a variety of common autoencoder setups and on VGG-9 with CIFAR-10. Through these experiments we demonstrate the interesting sensitivity that ADAM has to its momentum parameter $\beta_1$. We show that at very high values of the momentum parameter ($\beta_1 = 0.99$) ADAM outperforms a carefully tuned NAG on most of our experiments, in terms of getting lower training and test losses. On the other hand, NAG can sometimes do better when ADAM's $\beta_1$ is set to the most commonly used value: $\beta_1 = 0.9$, indicating the importance of tuning the hyperparameters of ADAM to get better generalization performance.

We also report experiments on different autoencoders to demonstrate that NAG has better abilities in terms of reducing the gradient norms, and it also produces iterates which exhibit an increasing trend for the minimum eigenvalue of the Hessian of the loss function at the iterates.

## 1 Introduction

Many optimization questions arising in machine learning can be cast as a finite sum optimization problem of the form: $\min_{\mathbf{x}} f(\mathbf{x})$ where $f(\mathbf{x}) = \frac{1}{k}\sum_{i=1}^{k} f_i(\mathbf{x})$. Most neural network problems also fall under a similar structure where each function $f_i$ is typically non-convex. A well-studied algorithm to solve such problems is Stochastic Gradient Descent (SGD), which uses updates of the form: $\mathbf{x}_{t+1} := \mathbf{x}_t - \alpha \nabla \tilde{f}_{i_t}(\mathbf{x}_t)$, where $\alpha$ is a step size, and $\tilde{f}_{i_t}$ is a function chosen randomly from $\{f_1, f_2, \ldots, f_k\}$ at time $t$. Often in neural networks, "momentum" is added to the SGD update to yield a two-step update process given as: $\mathbf{v}_{t+1} = \mu\mathbf{v}_t - \alpha\nabla\tilde{f}_{i_t}(\mathbf{x}_t)$ followed by $\mathbf{x}_{t+1} = \mathbf{x}_t + \mathbf{v}_{t+1}$. This algorithm is typically called the Heavy-Ball (HB) method (or sometimes classical momentum), with $\mu > 0$ called the momentum parameter (Polyak, 1987). In the context of neural nets, another variant of SGD that is popular is Nesterov's Accelerated Gradient (NAG), which can also be thought of as a momentum method (Sutskever et al., 2013), and has updates of the form $\mathbf{v}_{t+1} = \mu\mathbf{v}_t - \alpha\nabla\tilde{f}_{i_t}(\mathbf{x}_t + \mu\mathbf{v}_t)$ followed by $\mathbf{x}_{t+1} = \mathbf{x}_t + \mathbf{v}_{t+1}$ (see Algorithm 1 for more details).

Momentum methods like HB and NAG have been shown to have superior convergence properties compared to gradient descent in the deterministic setting both for convex and non-convex functions (Nesterov, 1983; Polyak, 1987; Zavriev & Kostyuk, 1993; Ochs, 2016; O'Neill & Wright, 2017; Jin et al., 2017). While (to the best of our knowledge) there is no clear theoretical justification in

the stochastic case of the benefits of NAG and HB over regular SGD in general (Yuan et al., 2016; Kidambi et al., 2018; Wiegerinck et al., 1994; Orr & Leen, 1994; Yang et al., 2016; Gadat et al., 2018), unless considering specialized function classes (Loizou & Richtárik, 2017); in practice, these momentum methods, and in particular NAG, have been repeatedly shown to have good convergence and generalization on a range of neural net problems (Sutskever et al., 2013; Lucas et al., 2018; Kidambi et al., 2018).

The performance of NAG (as well as HB and SGD), however, are typically quite sensitive to the selection of its hyper-parameters: step size, momentum and batch size (Sutskever et al., 2013). Thus, "adaptive gradient" algorithms such as RMSProp (Algorithm 2) (Tieleman & Hinton, 2012) and ADAM (Algorithm 3) (Kingma & Ba, 2014) have become very popular for optimizing deep neural networks (Melis et al., 2017; Xu et al., 2015; Denkowski & Neubig, 2017; Gregor et al., 2015; Radford et al., 2015; Bahar et al., 2017; Kiros et al., 2015). The reason for their widespread popularity seems to be the fact that they are believed to be easier to tune than SGD, NAG or HB. Adaptive gradient methods use as their update direction a vector which is the image under a linear transformation (often called the "diagonal pre-conditioner") constructed out of the history of the gradients, of a linear combination of all the gradients seen till now. It is generally believed that this "pre-conditioning" makes these algorithms much less sensitive to the selection of its hyper-parameters. A precursor to these RMSProp and ADAM was outlined in Duchi et al. (2011).

Despite their widespread use in neural net problems, adaptive gradients methods like RMSProp and ADAM lack theoretical justifications in the non-convex setting - even with exact/deterministic gradients (Bernstein et al., 2018). Further, there are also important motivations to study the behavior of these algorithms in the deterministic setting because of usecases where the amount of noise is controlled during optimization, either by using larger batches (Martens & Grosse, 2015; De et al., 2017; Babanezhad et al., 2015) or by employing variance-reducing techniques (Johnson & Zhang, 2013; Defazio et al., 2014).

Further, works like Wilson et al. (2017) and Keskar & Socher (2017) have shown cases where SGD (no momentum) and HB (classical momentum) generalize much better than RMSProp and ADAM with stochastic gradients. Wilson et al. (2017) also show that ADAM generalizes poorly for large enough nets and that RMSProp generalizes better than ADAM on a couple of neural network tasks (most notably in the character-level language modeling task). But in general it's not clear and no heuristics are known to the best of our knowledge to decide whether these insights about relative performances (generalization or training) between algorithms hold for other models or carry over to the full-batch setting.

**A summary of our contributions** In this work we try to shed some light on the above described open questions about adaptive gradient methods in the following two ways.

- To the best of our knowledge, this work gives the first convergence guarantees for adaptive gradient based standard neural-net training heuristics. Specifically we show run-time bounds for deterministic RMSProp and ADAM to reach approximate criticality on smooth non-convex functions, as well as for stochastic RMSProp under an additional assumption.

  Recently, Reddi et al. (2018) have shown in the setting of online convex optimization that there are certain sequences of convex functions where ADAM and RMSprop fail to converge to asymptotically zero average regret. We contrast our findings with Theorem 3 in Reddi et al. (2018). Their counterexample for ADAM is constructed in the stochastic optimization framework and is incomparable to our result about deterministic ADAM. Our proof of convergence to approximate critical points establishes a key conceptual point that for adaptive gradient algorithms one cannot transfer intuitions about convergence from online setups to their more common use case in offline setups.

- Our second contribution is empirical investigation into adaptive gradient methods, where our goals are different from what our theoretical results are probing. We test the convergence and generalization properties of RMSProp and ADAM and we compare their performance against NAG on a variety of autoencoder experiments on MNIST data, in both full and mini-batch settings. In the full-batch setting, we demonstrate that ADAM with very high values of the momentum parameter ($\beta_1 = 0.99$) matches or outperforms carefully tuned NAG and RMSProp, in terms of getting lower training and test losses. We show that as the autoencoder size keeps increasing, RMSProp fails to generalize pretty soon. In

the mini-batch experiments we see exactly the same behaviour for large enough nets. We further validate this behavior on an image classification task on CIFAR-10 using a VGG-9 convolutional neural network, the results to which we present in the Appendix E.

We note that recently it has been shown by Lucas et al. (2018), that there are problems where NAG generalizes better than ADAM even after tuning $\beta_1$. In contrast our experiments reveal controlled setups where tuning ADAM's $\beta_1$ closer to 1 than usual practice helps close the generalization gap with NAG and HB which exists at standard values of $\beta_1$.

**Remark.** Much after this work was completed we came to know of a related paper (Li & Orabona, 2018) which analyzes convergence rates of a modification of AdaGrad (not RMSPRop or ADAM). After the initial version of our work was made public, a few other analysis of adaptive gradient methods have also appeared like Chen et al. (2018), Zhou et al. (2018) and Zaheer et al. (2018).

## 2 NOTATIONS AND PSEUDOCODES

Firstly we define the smoothness property that we assume in our proofs for all our non-convex objectives. This is a standard assumption used in the optimization literature.

**Definition 1.** $L-$**smoothness** If $f : \mathbb{R}^d \to \mathbb{R}$ is at least once differentiable then we call it $L-$smooth for some $L > 0$ if for all $\mathbf{x}, \mathbf{y} \in \mathbb{R}^d$ the following inequality holds,

$$f(\mathbf{y}) \leq f(\mathbf{x}) + \langle \nabla f(\mathbf{x}), \mathbf{y} - \mathbf{x} \rangle + \frac{L}{2} \left\| \mathbf{y} - \mathbf{x} \right\|^2.$$

We need one more definition that of square-root of diagonal matrices,

**Definition 2. Square root of the Penrose inverse** If $\mathbf{v} \in \mathbb{R}^d$ and $V = \text{diag}(\mathbf{v})$ then we define, $V^{-\frac{1}{2}} := \sum_{i \in \text{Support}(\mathbf{v})} \frac{1}{\sqrt{\mathbf{v}_i}} \mathbf{e}_i \mathbf{e}_i^T$, where $\{\mathbf{e}_i\}_{\{i=1,\ldots,d\}}$ is the standard basis of $\mathbb{R}^d$

Now we list out the pseudocodes used for NAG, RMSProp and ADAM in theory and experiments,

**Nesterov's Accelerated Gradient (NAG) Algorithm**

---
**Algorithm 1** NAG
---
1: **Input :** A step size $\alpha$, momentum $\mu \in [0, 1)$, and an initial starting point $\mathbf{x}_1 \in \mathbb{R}^d$, and we are given query access to a (possibly noisy) oracle for gradients of $f : \mathbb{R}^d \to \mathbb{R}$.
2: **function** NAG($\mathbf{x}_1, \alpha, \mu$)
3:     **Initialize :** $\mathbf{v}_1 = \mathbf{0}$
4:     **for** $t = 1, 2, \ldots$ **do**
5:         $\mathbf{v}_{t+1} = \mu \mathbf{v}_t + \nabla f(\mathbf{x}_t)$
6:         $\mathbf{x}_{t+1} = \mathbf{x}_t - \alpha(\nabla f(\mathbf{x}_t) + \mu \mathbf{v}_{t+1})$
7:     **end for**
8: **end function**

---

**RMSProp Algorithm**

---
**Algorithm 2** RMSProp
---
1: **Input :** A constant vector $\mathbb{R}^d \ni \xi \mathbf{1}_d \geq 0$, parameter $\beta_2 \in [0, 1)$, step size $\alpha$, initial starting point $\mathbf{x}_1 \in \mathbb{R}^d$, and we are given query access to a (possibly noisy) oracle for gradients of $f : \mathbb{R}^d \to \mathbb{R}$.
2: **function** RMSPROP($\mathbf{x}_1, \beta_2, \alpha, \xi$)
3:     **Initialize :** $\mathbf{v}_0 = \mathbf{0}$
4:     **for** $t = 1, 2, \ldots$ **do**
5:         $\mathbf{g}_t = \nabla f(\mathbf{x}_t)$
6:         $\mathbf{v}_t = \beta_2 \mathbf{v}_{t-1} + (1 - \beta_2)(\mathbf{g}_t^2 + \xi \mathbf{1}_d)$
7:         $V_t = \text{diag}(\mathbf{v}_t)$
8:         $\mathbf{x}_{t+1} = \mathbf{x}_t - \alpha V_t^{-\frac{1}{2}} \mathbf{g}_t$
9:     **end for**
10: **end function**

---

**ADAM Algorithm**

---

**Algorithm 3** ADAM

---

1: **Input :** A constant vector $\mathbb{R}^d \ni \xi\mathbf{1}_d > 0$, parameters $\beta_1, \beta_2 \in [0,1)$, a sequence of step sizes $\{\alpha_t\}_{t=1,2..}$, initial starting point $\mathbf{x}_1 \in \mathbb{R}^d$, and we are given oracle access to the gradients of $f : \mathbb{R}^d \to \mathbb{R}$.

2: **function** ADAM($\mathbf{x}_1, \beta_1, \beta_2, \alpha, \xi$)

3:     **Initialize :**  $\mathbf{m}_0 = \mathbf{0}$,  $\mathbf{v}_0 = \mathbf{0}$

4:     **for** $t = 1, 2, \ldots$ **do**

5:         $\mathbf{g}_t = \nabla f(\mathbf{x}_t)$

6:         $\mathbf{m}_t = \beta_1 \mathbf{m}_{t-1} + (1-\beta_1)\mathbf{g}_t$

7:         $\mathbf{v}_t = \beta_2 \mathbf{v}_{t-1} + (1-\beta_2)\mathbf{g}_t^2$

8:         $V_t = \text{diag}(\mathbf{v}_t)$

9:         $\mathbf{x}_{t+1} = \mathbf{x}_t - \alpha_t \left( V_t^{\frac{1}{2}} + \text{diag}(\xi\mathbf{1}_d) \right)^{-1} \mathbf{m}_t$

10:    **end for**

11: **end function**

---

## 3   Convergence Guarantees for ADAM and RMSProp

Previously it has been shown in Rangamani et al. (2017) that mini-batch RMSProp can off-the-shelf do autoencoding on depth 2 autoencoders trained on MNIST data while similar results using non-adaptive gradient descent methods requires much tuning of the step-size schedule. Here we give the first result about convergence to criticality for stochastic RMSProp albeit under a certain technical assumption about the training set (and hence on the first order oracle). Towards that we need the following definition,

**Definition 3. The sign function**
We define the function sign : $\mathbb{R}^d \to \{-1, 1\}^d$ s.t it maps $\mathbf{v} \mapsto (1 \text{ if } \mathbf{v}_i \geq 0 \text{ else } -1)_{i=1,\ldots,d}$.

**Theorem 3.1. Stochastic RMSProp converges to criticality (Proof in subsection A.1)** Let $f : \mathbb{R}^d \to \mathbb{R}$ be $L-$smooth and be of the form $f = \frac{1}{k}\sum_{i=1}^k f_i$ s.t. (a) each $f_i$ is at least once differentiable, (b) the gradients are s.t $\forall \mathbf{x} \in \mathbb{R}^d, \forall p, q \in \{1, \ldots, k\}, \text{sign}(\nabla f_p(\mathbf{x})) = \text{sign}(\nabla f_q(\mathbf{x}))$, (c) $\sigma_f < \infty$ is an upperbound on the norm of the gradients of $f_i$ and (d) $f$ has a minimizer, i.e., there exists $\mathbf{x}_*$ such that $f(\mathbf{x}_*) = \min_{\mathbf{x} \in \mathbb{R}^d} f(\mathbf{x})$. Let the gradient oracle be s.t when invoked at some $\mathbf{x}_t \in \mathbb{R}^d$ it uniformly at random picks $i_t \sim \{1, 2, .., k\}$ and returns, $\nabla f_{i_t}(\mathbf{x}_t) = \mathbf{g}_t$. Then corresponding to any $\epsilon, \xi > 0$ and a starting point $\mathbf{x}_1$ for Algorithm 2, we can define, $T \leq \frac{1}{\epsilon^4}\left( \frac{2L\sigma_f^2(\sigma_f^2 + \xi)(f(\mathbf{x}_1) - f(\mathbf{x}_*))}{(1-\beta_2)\xi} \right)$ s.t. we are guaranteed that the iterates of Algorithm 2 using a constant step-length of, $\alpha = \frac{1}{\sqrt{T}}\sqrt{\frac{2\xi(1-\beta_2)(f(\mathbf{x}_1) - f(\mathbf{x}_*))}{\sigma_f^2 L}}$ will find an $\epsilon-$critical point in at most $T$ steps in the sense that, $\min_{t=1,2\ldots,T}\mathbb{E}[\|\nabla f(\mathbf{x}_t)\|^2] \leq \epsilon^2$.     □

**Remark.** We note that the theorem above continues to hold even if the constraint $(b)$ that we have about the signs of the gradients of the $\{f_i\}_{i=1,\ldots,k}$ *only* holds on the points in $\mathbb{R}^d$ that the stochastic RMSProp visits and its not necessary for the constraint to be true everywhere in the domain. Further we can say in otherwords that this constraint ensures all the options for the gradient that this stochastic oracle has at any point, to lie in the same orthant of $\mathbb{R}^d$ though this orthant itself can change from one iterate of the next. A related result was concurrently shown by Zaheer et al. (2018).

Next we see that such sign conditions are not necessary to guarantee convergence of the deterministic RMSProp which corresponds to the full-batch RMSProp experiments in Section 5.3.

**Theorem 3.2. Convergence of deterministic RMSProp - the version with standard speeds (Proof in subsection A.2)** Let $f : \mathbb{R}^d \to \mathbb{R}$ be $L-$smooth and let $\sigma < \infty$ be an upperbound on the norm of the gradient of $f$. Assume also that $f$ has a minimizer, i.e., there exists $\mathbf{x}_*$ such that $f(\mathbf{x}_*) = \min_{\mathbf{x} \in \mathbb{R}^d} f(\mathbf{x})$. Then the following holds for Algorithm 2 with a deterministic gradient oracle:

For any $\epsilon, \xi > 0$, using a constant step length of $\alpha_t = \alpha = \frac{(1-\beta_2)\xi}{L\sqrt{\sigma^2+\xi}}$ for $t = 1, 2, ...$, guarantees that $\|\nabla f(\mathbf{x}_t)\| \leq \epsilon$ for some $t \leq \frac{1}{\epsilon^2} \times \frac{2L(\sigma^2+\xi)(f(\mathbf{x}_1)-f(\mathbf{x}_*))}{(1-\beta_2)\xi}$, where $\mathbf{x}_1$ is the first iterate of the algorithm. □

One might wonder if the $\xi$ parameter introduced in the algorithms above is necessary to get convergence guarantees for RMSProp. Towards that in the following theorem we show convergence of another variant of deterministic RMSProp which does not use the $\xi$ parameter and instead uses other assumptions on the objective function and step size modulation. But these tweaks to eliminate the need of $\xi$ come at the cost of the convergence rates getting weaker.

**Theorem 3.3. Convergence of deterministic RMSProp - the version with no $\xi$ shift (Proof in subsection A.3)** Let $f : \mathbb{R}^d \to \mathbb{R}$ be $L-$smooth and let $\sigma < \infty$ be an upperbound on the norm of the gradient of $f$. Assume also that $f$ has a minimizer, i.e., there exists $\mathbf{x}_*$ such that $f(\mathbf{x}_*) = \min_{\mathbf{x} \in \mathbb{R}^d} f(\mathbf{x})$, and the function $f$ be bounded from above and below by constants $B_\ell$ and $B_u$ as $B_l \leq f(\mathbf{x}) \leq B_u$ for all $\mathbf{x} \in \mathbb{R}^d$. Then for any $\epsilon > 0$, $\exists T = O(\frac{1}{\epsilon^4})$ s.t. the Algorithm 2 with a deterministic gradient oracle and $\xi = 0$ is guaranteed to reach a $t$-th iterate s.t. $1 \leq t \leq T$ and $\|\nabla f(\mathbf{x}_t)\| \leq \epsilon$. □

In Section 5.3 we show results of our experiments with full-batch ADAM. Towards that, we analyze deterministic ADAM albeit in the small $\beta_1$ regime. We note that a small $\beta_1$ does not cut-off contributions to the update direction from gradients in the arbitrarily far past (which are typically significantly large), and neither does it affect the non-triviality of the pre-conditioner which does not depend on $\beta_1$ at all.

**Theorem 3.4. Deterministic ADAM converges to criticality (Proof in subsection A.4)** Let $f : \mathbb{R}^d \to \mathbb{R}$ be $L-$smooth and let $\sigma < \infty$ be an upperbound on the norm of the gradient of $f$. Assume also that $f$ has a minimizer, i.e., there exists $\mathbf{x}_*$ such that $f(\mathbf{x}_*) = \min_{\mathbf{x} \in \mathbb{R}^d} f(\mathbf{x})$. Then the following holds for Algorithm 3:

For any $\epsilon > 0$, $\beta_1 < \frac{\epsilon}{\epsilon+\sigma}$ and $\xi > \frac{\sigma^2\beta_1}{-\beta_1\sigma+\epsilon(1-\beta_1)}$, there exist step sizes $\alpha_t > 0$, $t = 1, 2, \ldots$ and a natural number $T$ (depending on $\beta_1, \xi$) such that $\|\nabla f(\mathbf{x}_t)\| \leq \epsilon$ for some $t \leq T$.

In particular if one sets $\beta_1 = \frac{\epsilon}{\epsilon+2\sigma}$, $\xi = 2\sigma$, and $\alpha_t = \frac{\|\mathbf{g}_t\|^2}{L(1-\beta_1^t)^2} \frac{4\epsilon}{3(\epsilon+2\sigma)^2}$ where $\mathbf{g}_t$ is the gradient of the objective at the $t^{th}$ iterate, then $T$ can be taken to be $\frac{9L\sigma^2}{\epsilon^6}[f(\mathbf{x}_2) - f(\mathbf{x}_*)]$, where $\mathbf{x}_2$ is the second iterate of the algorithm. □

Our motivations towards the above theorem were primarily rooted in trying to understand the situations where ADAM can converge at all (given the negative results about ADAM as in Reddi et al. (2018)). But we point out that it remains open to tighten the analysis of deterministic ADAM and obtain faster rates than what we have shown in the theorem above.

**Remark.** It is sometimes believed that ADAM gains over RMSProp because of its "bias correction term" which refers to the step length of ADAM having an iteration dependence of the following form, $\sqrt{1-\beta_2^t}/(1-\beta_1^t)$. In the above theorem, we note that the $1/(1-\beta_1^t)$ term of this "bias correction term" naturally comes out from theory!

# 4 EXPERIMENTAL SETUP

For testing the empirical performance of ADAM and RMSProp, we perform experiments on fully connected autoencoders using ReLU activations and shared weights and on CIFAR-10 using VGG-9, a convolutional neural network. Let $\mathbf{z} \in \mathbb{R}^d$ be the input vector to the autoencoder, $\{W_i\}_{i=1,..,\ell}$ denote the weight matrices of the net and $\{\mathbf{b}_i\}_{i=1,...,2\ell}$ be the bias vectors. Then the output $\hat{\mathbf{z}} \in \mathbb{R}^d$ of the autoencoder is defined as $\hat{\mathbf{z}} = W_1^\top \sigma(\ldots \sigma(W_{\ell-1}^\top \sigma(W_\ell^\top \mathbf{a} + \mathbf{b}_{\ell+1}) + \mathbf{b}_{\ell+2}) \ldots) + \mathbf{b}_{2\ell}$ where $\mathbf{a} = \sigma(W_\ell \sigma(\ldots \sigma(W_2 \sigma(W_1 \mathbf{z} + \mathbf{b}_1) + \mathbf{b}_2) \ldots) + \mathbf{b}_\ell)$. This defines an autoencoder with $2\ell - 1$ hidden layers using $\ell$ weight matrices and $2\ell$ bias vectors. Thus, the parameters of this model are given by $\mathbf{x} = [\text{vec}(W_1)^\top ... \text{vec}(W_\ell)^\top \mathbf{b}_1^\top ... \mathbf{b}_{2\ell}^\top]^\top$ (where we imagine all vectors to be column vectors by default). The loss function, for an input $\mathbf{z}$ is then given by: $f(\mathbf{z}; \mathbf{x}) = \|\mathbf{z} - \hat{\mathbf{z}}\|^2$.

Such autoencoders are a fairly standard setup that have been used in previous work (Arpit et al., 2015; Baldi, 2012; Kuchaiev & Ginsburg, 2017; Vincent et al., 2010). There have been relatively

fewer comparisons of ADAM and RMSProp with other methods on a regression setting. We were motivated by Rangamani et al. (2017) who had undertaken a theoretical analysis of autoencoders and in their experiments had found RMSProp to have good reconstruction error for MNIST when used on even just 2 layer ReLU autoencoders.

To keep our experiments as controlled as possible, we make all layers in a network have the same width (which we denote as $h$). Thus, given a size $d$ for the input image, the weight matrices (as defined above) are given by: $W_1 \in \mathbb{R}^{h \times d}$, $W_i \in \mathbb{R}^{h \times h}$, $i = 2, \ldots, \ell$. This allowed us to study the effect of increasing depth $\ell$ or width $h$ without having to deal with added confounding factors. For all experiments, we use the standard "Glorot initialization" for the weights (Glorot & Bengio, 2010), where each element in the weight matrix is initialized by sampling from a uniform distribution with $[-\text{limit}, \text{limit}]$, $\text{limit} = \sqrt{6/(\text{fan}_{in} + \text{fan}_{out})}$, where $\text{fan}_{in}$ denotes the number of input units in the weight matrix, and $\text{fan}_{out}$ denotes the number of output units in the weight matrix. All bias vectors were initialized to zero. No regularization was used.

We performed autoencoder experiments on the MNIST dataset for various network sizes (i.e., different values of $\ell$ and $h$). We implemented all experiments using TensorFlow (Abadi et al., 2016) using an NVIDIA GeForce GTX 1080 Ti graphics card. We compared the performance of ADAM and RMSProp with Nesterov's Accelerated Gradient (NAG). All experiments were run for $10^5$ iterations. We tune over the hyper-parameters for each optimization algorithm using a grid search as described in the Appendix (Section B). To pick the best set of hyper-parameters, we choose the ones corresponding to the lowest loss on the training set at the end of $10^5$ iterations. Further, to cut down on the computation time so that we can test a number of different neural net architectures, we crop the MNIST image from $28 \times 28$ down to a $22 \times 22$ image by removing 3 pixels from each side (almost all of which is whitespace).

**Full-batch experiments** We are interested in first comparing these algorithms in the full-batch setting. To do this in a computationally feasible way, we consider a subset of the MNIST dataset (we call this: mini-MNIST), which we build by extracting the first 5500 images in the training set and first 1000 images in the test set in MNIST. Thus, the training and testing datasets in mini-MNIST is 10% of the size of the MNIST dataset. Thus the training set in mini-MNIST contains 5500 images, while the test set contains 1000 images. This subset of the dataset is a fairly reasonable approximation of the full MNIST dataset (i.e., contains roughly the same distribution of labels as in the full MNIST dataset), and thus a legitimate dataset to optimize on.

**Mini-batch experiments** To test if our conclusions on the full-batch case extend to the mini-batch case, we then perform the same experiments in a mini-batch setup where we fix the mini-batch size at 100. For the mini-batch experiment, we consider the full training set of MNIST, instead of the mini-MNIST dataset considered for the full-batch experiments and we also test on CIFAR-10 using VGG-9, a convolutional neural network.

## 5 EXPERIMENTAL RESULTS

### 5.1 RMSPROP AND ADAM ARE SENSITIVE TO CHOICE OF $\xi$

The $\xi$ parameter is a feature of the default implementations of RMSProp and ADAM such as in TensorFlow. Most interestingly this strictly positive parameter is crucial for our proofs. In this section we present experimental evidence that attempts to clarify that this isn't merely a theoretical artefact but its value indeed has visible effect on the behaviours of these algorithms. We see in Figure 1 that on increasing the value of this fixed shift parameter $\xi$, ADAM in particular, is strongly helped towards getting lower gradient norms and lower test losses though it can hurt its ability to get lower training losses. The plots are shown for optimally tuned values for the other hyper-parameters.

### 5.2 TRACKING $\lambda_{min}$(HESSIAN) OF THE LOSS FUNCTION

To check whether NAG, ADAM or RMSProp is capable of consistently moving from a "bad" saddle point to a "good" saddle point region, we track the most negative eigenvalue of the Hessian $\lambda_{min}(Hessian)$. Even for a very small neural network with around $10^5$ parameters, it is still intractable to store the full Hessian matrix in memory to compute the eigenvalues. Instead, we use the

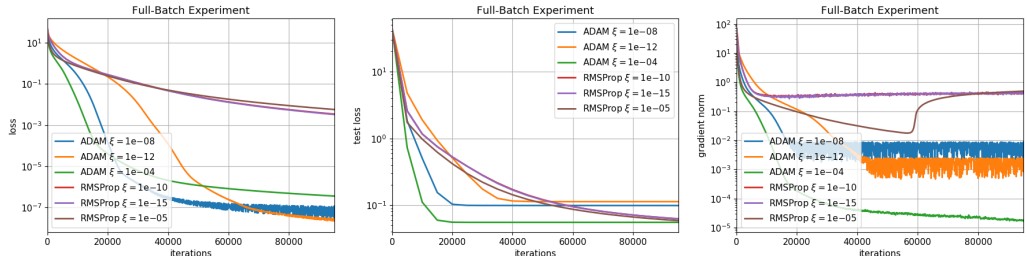

Figure 1: Optimally tuned parameters for different $\xi$ values. 1 hidden layer network of 1000 nodes; *Left*: Loss on training set; *Middle*: Loss on test set; *Right*: Gradient norm on training set

Scipy library function `scipy.sparse.linalg.eigsh` that can use a function that computes the matrix-vector products to compute the eigenvalues of the matrix (Lehoucq et al., 1998). Thus, for finding the eigenvalues of the Hessian, it is sufficient to be able to do Hessian-vector products. This can be done exactly in a fairly efficient way (Townsend, 2008).

We display a representative plot in Figure 2 which shows that NAG in particular has a distinct ability to gradually, but consistently, keep increasing the minimum eigenvalue of the Hessian while continuing to decrease the gradient norm. However unlike as in deeper autoencoders in this case the gradient norms are consistently bigger for NAG, compared to RMSProp and ADAM. In contrast, RSMProp and ADAM quickly get to a high value of the minimum eigenvalue and a small gradient norm, but somewhat stagnate there. In short, the trend looks better for NAG, but in actual numbers RMSProp and ADAM do better.

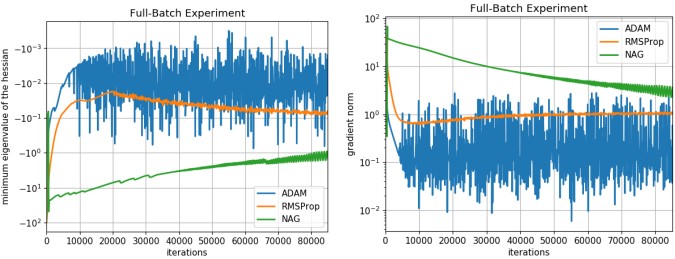

Figure 2: Tracking the smallest eigenvalue of the Hessian on a 1 hidden layer network of size 300. *Left*: Minimum Hessian eigenvalue. *Right*: Gradient norm on training set.

### 5.3 Comparing performance in the full-batch setting

In Figure 3, we show how the training loss, test loss and gradient norms vary through the iterations for RMSProp, ADAM (at $\beta_1 = 0.9$ and 0.99) and NAG (at $\mu = 0.9$ and 0.99) on a 3 hidden layer autoencoder with 1000 nodes in each hidden layer trained on mini-MNIST. Appendix D.1 and D.2 have more such comparisons for various neural net architectures with varying depth and width and input image sizes, where the following qualitative results also extend.

**Conclusions from the full-batch experiments of training autoencoders on mini-MNIST:**

- Pushing $\beta_1$ closer to 1 significantly helps ADAM in getting lower training and test losses and at these values of $\beta_1$, it has better performance on these metrics than all the other algorithms. One sees cases like the one displayed in Figure 3 where ADAM at $\beta_1 = 0.9$ was getting comparable or slightly worse test and training errors than NAG. But once $\beta_1$ gets closer to 1, ADAM's performance sharply improves and gets better than other algorithms.

- Increasing momentum helps NAG get lower gradient norms though on larger nets it might hurt its training or test performance. NAG does seem to get the lowest gradient norms compared to the other algorithms, except for single hidden layer networks like in Figure 2.

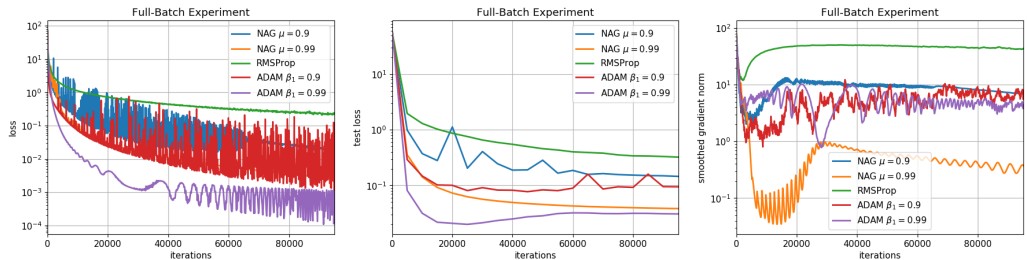

Figure 3: Full-batch experiments on a 3 hidden layer network with 1000 nodes in each layer; *Left*: Loss on training set; *Middle*: Loss on test set; *Right*: Gradient norm on training set

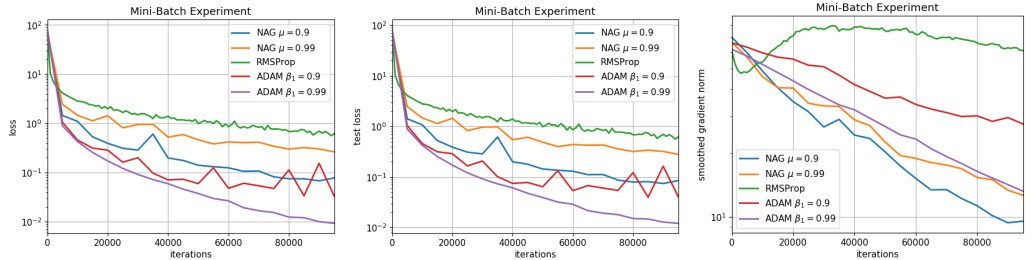

Figure 4: Mini-batch experiments on a network with 5 hidden layers of 1000 nodes each; *Left*: Loss on training set; *Middle*: Loss on test set; *Right*: Gradient norm on training set

### 5.4 CORROBORATING THE FULL-BATCH BEHAVIORS IN THE MINI-BATCH SETTING

In Figure 4, we show how training loss, test loss and gradient norms vary when using mini-batches of size 100, on a 5 hidden layer autoencoder with 1000 nodes in each hidden layer trained on the full MNIST dataset. The same phenomenon as here has been demonstrated in more such mini-batch comparisons on autoencoder architectures with varying depths and widths in Appendix D.3 and on VGG-9 with CIFAR-10 in Appendix E.

**Conclusions from the mini-batch experiments of training autoencoders on full MNIST dataset:**

- Mini-batching does seem to help NAG do better than ADAM on small nets. However, for larger nets, the full-batch behavior continues, i.e., when ADAM's momentum parameter $\beta_1$ is pushed closer to 1, it gets better generalization (significantly lower test losses) than NAG at any momentum tested.

- In general, for all metrics (test loss, training loss and gradient norm reduction) both ADAM as well as NAG seem to improve in performance when their momentum parameter ($\mu$ for NAG and $\beta_1$ for ADAM) is pushed closer to 1. This effect, which was present in the full-batch setting, seems to get more pronounced here.

- As in the full-batch experiments, NAG continues to have the best ability to reduce gradient norms while for larger enough nets, ADAM at large momentum continues to have the best training error.

## 6 CONCLUSION

To the best of our knowledge, we present the first theoretical guarantees of convergence to criticality for the immensely popular algorithms RMSProp and ADAM in their most commonly used setting of optimizing a non-convex objective.

By our experiments, we have sought to shed light on the important topic of the interplay between adaptivity and momentum in training nets. By choosing to study textbook autoencoder architectures

where various parameters of the net can be changed controllably we highlight the following two aspects that (a) the value of the gradient shifting hyperparameter $\xi$ has a significant influence on the performance of ADAM and RMSProp and (b) ADAM seems to perform particularly well (often supersedes Nesterov accelerated gradient method) when its momentum parameter $\beta_1$ is very close to 1. On VGG-9 with CIFAR-10 and for the task of training autoencoders on MNIST we have verified these conclusions across different widths and depths of nets as well as in the full-batch and the mini-batch setting (with large nets) and under compression of the input/out image size. Curiously enough, this regime of $\beta_1$ being close to 1 is currently not within the reach of our proof techniques of showing convergence for ADAM. Our experiments give strong reasons to try to advance theory in this direction in future work.

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

# Appendix

## A Proofs of convergence of (stochastic) RMSProp and Adam

### A.1 Proving stochastic RMSProp (Proof of Theorem 3.1)

Now we give the proof of Theorem 3.1.

*Proof.* We define $\sigma_t := \max_{k=1,..,t} \|\nabla f_{i_k}(\mathbf{x}_k)\|$ and we solve the recursion for $\mathbf{v}_t$ as, $\mathbf{v}_t = (1 - \beta_2) \sum_{k=1}^{t} \beta_2^{t-k}(\mathbf{g}_k^2 + \xi)$. This lets us write the following bounds,

$$\lambda_{min}(V_t^{-\frac{1}{2}}) \geq \frac{1}{\sqrt{\max_{i=1,..,d}(\mathbf{v}_t)_i}} \geq \frac{1}{\sqrt{\max_{i=1,..,d}((1-\beta_2)\sum_{k=1}^{t}\beta_2^{t-k}(\mathbf{g}_k^2 + \xi\mathbf{1}_d)_i)}}$$

$$\geq \frac{1}{\sqrt{1-\beta_2^t}\sqrt{\sigma_t^2 + \xi}}$$

Now we define, $\epsilon_t := \min_{k=1,..,t, i=1,..,d}(\nabla f_{i_k}(\mathbf{x}_k))_i^2$ and this lets us get the following bounds,

$$\lambda_{max}(V_t^{-\frac{1}{2}}) \leq \frac{1}{\min_{i=1,..,d}(\sqrt{(\mathbf{v}_t)_i})} \leq \frac{1}{\sqrt{(1-\beta_2^t)}\sqrt{(\xi + \epsilon_t)}}$$

Now we invoke the bounded gradient assumption about the $f_i$ functions and replace in the above equation the eigenvalue bounds of the pre-conditioner by worst-case estimates $\mu_{\max}$ and $\mu_{\min}$ defined as,

$$\lambda_{min}(V_t^{-\frac{1}{2}}) \geq \frac{1}{\sqrt{\sigma_f^2 + \xi}} := \mu_{\min}$$

$$\lambda_{max}(V_t^{-\frac{1}{2}}) \leq \frac{1}{\sqrt{(1-\beta_2)}\sqrt{\xi}} := \mu_{max}$$

Using the $L$-smoothness of $f$ between consecutive iterates $\mathbf{x}_t$ and $\mathbf{x}_{t+1}$ we have,

$$f(\mathbf{x}_{t+1}) \leq f(\mathbf{x}_t) + \langle \nabla f(\mathbf{x}_t), \mathbf{x}_{t+1} - \mathbf{x}_t \rangle + \frac{L}{2}\|\mathbf{x}_{t+1} - \mathbf{x}_t\|^2$$

We note that the update step of stochastic RMSProp is $x_{t+1} = x_t - \alpha(V_t)^{-\frac{1}{2}}g_t$ where $g_t$ is the stochastic gradient at iterate $x_t$. Let $H_t = \{\mathbf{x}_1, \mathbf{x}_2, .., \mathbf{x}_t\}$ be the set of random variables corresponding to the first $t$ iterates. The assumptions we have about the stochastic oracle give us the following relations, $\mathbb{E}[g_t] = \nabla f(x_t)$ and $\mathbb{E}[\|g_t\|^2] \leq \sigma_f^2$. Now we can invoke these stochastic oracle's properties and take a conditional (on $H_t$) expectation over $\mathbf{g}_t$ of the $L-$smoothness in equation to get,

$$\mathbb{E}[f(\mathbf{x}_{t+1}) \mid H_t] \leq f(\mathbf{x}_t) - \alpha\mathbb{E}\left[\langle \nabla f(\mathbf{x}_t), (V_t)^{-\frac{1}{2}}\mathbf{g}_t \rangle \mid H_t\right] + \frac{\alpha^2 L}{2}\mathbb{E}\left[\|(V_t)^{-\frac{1}{2}}\mathbf{g}_t\|^2 \mid H_t\right]$$

$$\leq f(\mathbf{x}_t) - \alpha\mathbb{E}\left[\langle \nabla f(\mathbf{x}_t), (V_t)^{-\frac{1}{2}}\mathbf{g}_t \rangle \mid H_t\right] + \mu_{\max}^2\frac{\alpha^2 L}{2}\mathbb{E}\left[\|\mathbf{g}_t\|^2 \mid H_t\right]$$

$$\leq f(\mathbf{x}_t) - \alpha\mathbb{E}\left[\langle \nabla f(\mathbf{x}_t), (V_t)^{-\frac{1}{2}}\mathbf{g}_t \rangle \mid H_t\right] + \mu_{\max}^2\frac{\alpha^2 \sigma_f^2 L}{2} \tag{1}$$

We now separately analyze the middle term in the RHS above in Lemma A.1 below and we get,

$$\mathbb{E}[\langle \nabla f(\mathbf{x}_t), (V_t)^{-\frac{1}{2}}\mathbf{g}_t \rangle \mid H_t] \geq \mu_{min}\|\nabla f(\mathbf{x}_t)\|^2$$

We substitute the above into equation 1 and take expectations over $H_t$ to get,

$$\mathbb{E}[f(\mathbf{x}_{t+1}) - f(\mathbf{x}_t)] \leq -\alpha\mu_{min}\|\nabla f(\mathbf{x}_t)\|^2 + \mu_{max}^2 \frac{\alpha^2\sigma_f^2 L}{2}$$

$$\implies \mathbb{E}[\|\nabla f(\mathbf{x}_t)\|^2] \leq \frac{1}{\alpha\mu_{min}}\mathbb{E}[f(\mathbf{x}_t) - f(\mathbf{x}_{t+1})] + \frac{\alpha\sigma_f^2 L}{2}\frac{\mu_{max}^2}{\mu_{min}} \quad (2)$$

Doing the above replacements to upperbound the RHS of equation 2 and summing the inequation over $t = 1$ to $t = T$ and taking the average and replacing the LHS by a lowerbound of it, we get,

$$\min_{t=1...T}\mathbb{E}[\|\nabla f(\mathbf{x}_t)\|^2] \leq \frac{1}{\alpha T\mu_{\min}}\mathbb{E}[f(\mathbf{x}_1) - f(\mathbf{x}_{T+1})] + \frac{\alpha\sigma_f^2 L}{2}\frac{\mu_{\max}^2}{\mu_{\min}}$$

$$\leq \frac{1}{\alpha T\mu_{\min}}(f(\mathbf{x}_1) - f(\mathbf{x}_*)) + \frac{\alpha\sigma_f^2 L}{2}\frac{\mu_{\max}^2}{\mu_{\min}}$$

Replacing into the RHS above the optimal choice of,

$$\alpha = \frac{1}{\sqrt{T}}\sqrt{\frac{2(f(\mathbf{x}_1) - f(\mathbf{x}_*))}{\sigma_f^2 L\mu_{\max}^2}} = \frac{1}{\sqrt{T}}\sqrt{\frac{2\xi(1-\beta_2)(f(\mathbf{x}_1) - f(\mathbf{x}_*))}{\sigma_f^2 L}}$$

we get,

$$\min_{t=1...T}\mathbb{E}[\|\nabla f(\mathbf{x}_t)\|^2] \leq 2\sqrt{\frac{1}{T\mu_{\min}}(f(\mathbf{x}_1) - f(\mathbf{x}_*)) \times \frac{L\sigma_f^2}{2}\frac{\mu_{\max}^2}{\mu_{\min}}} = \frac{1}{\sqrt{T}}\sqrt{\frac{2L\sigma_f^2(\sigma_f^2 + \xi)(f(\mathbf{x}_1) - f(\mathbf{x}_*))}{(1-\beta_2)\xi}}$$

Thus stochastic RMSProp with the above step-length is guaranteed is reach $\epsilon-$criticality in number of iterations given by, $T \leq \frac{1}{\epsilon^4}\left(\frac{2L\sigma_f^2(\sigma_f^2+\xi)(f(\mathbf{x}_1)-f(\mathbf{x}_*))}{(1-\beta_2)\xi}\right)$ □

**Lemma A.1.** At any time $t$, the following holds,

$$\mathbb{E}[\langle \nabla f(x_t), V_t^{-1/2}g_t \rangle \mid H_t] \geq \mu_{\min}\|\nabla f(x_t)\|^2$$

*Proof.*

$$\mathbb{E}\left[\langle \nabla f(x_t), V_t^{-1/2}g_t \rangle \mid H_t\right] = \mathbb{E}\left[\sum_{i=1}^{d}\nabla_i f(x_t)(V_t^{-1/2})_{ii}(g_t)_i \mid H_t\right]$$

$$= \sum_{i=1}^{d}\nabla_i f(x_t)\mathbb{E}\left[(V_t^{-1/2})_{ii}(g_t)_i \mid H_t\right] \quad (3)$$

Now we introduce some new variables to make the analysis easier to present. Let $a_{pi} := [\nabla f_p(\mathbf{x}_t)]_i$ where $p$ indexes the training data set, $p \in \{1, \ldots, k\}$. (conditioned on $H_t$, $a_{pi}$s are constants) This implies, $\nabla_i f(x_t) = \frac{1}{k}\sum_{p=1}^{k}a_{pi}$ We recall that $\mathbb{E}[(\mathbf{g}_t)_i] = \nabla_i f(\mathbf{x}_t)$ where the expectation is taken over the oracle call at the $t^{th}$ update step. Further our instantiation of the oracle is equivalent to doing the uniformly at random sampling, $(\mathbf{g}_t)_i \sim \{a_{pi}\}_{p=1,\ldots,k}$.

Given that we have, $V_t = \text{diag}(\mathbf{v}_t)$ with $\mathbf{v}_t = (1 - \beta_2)\sum_{k=1}^{t}\beta_2^{t-k}(\mathbf{g}_k^2 + \xi\mathbf{1}_d)$ this implies, $(V_t^{-1/2})_{ii} = \frac{1}{\sqrt{(1-\beta_2)(\mathbf{g}_t)_i^2 + d_i}}$ where we have defined $d_i := (1 - \beta_2)\xi + (1 - \beta_2)\sum_{k=1}^{t-1}\beta_2^{t-k}((\mathbf{g}_k)_i^2 + \xi)$. (conditioned on $H_t$, $d_i$ is a constant) This leads to an explicit form of the needed expectation over the $t^{th}-$oracle call as,

$$\mathbb{E}\left[(V_t^{-1/2})_{ii}(g_t)_i \mid H_t\right] = \mathbb{E}\left[(V_t^{-1/2})_{ii}(g_t)_i \mid H_t\right]$$

$$= \mathbb{E}_{(\mathbf{g}_t)_i \sim \{a_{pi}\}_{p=1,\dots,k}}\left[\frac{(g_t)_i}{\sqrt{(1-\beta_2)(g_t)_i^2 + d_i}} \mid H_t\right]$$

$$= \frac{1}{k}\sum_{p=1}^{k}\frac{a_{pi}}{\sqrt{(1-\beta_2)a_{pi}^2 + d_i}}$$

Substituting the above (and the definition of the constants $a_{pi}$) back into equation 3 we have,

$$\mathbb{E}\left[\langle \nabla f(x_t), V_t^{-1/2}g_t\rangle \mid H_t\right] = \sum_{i=1}^{d}\left(\frac{1}{k}\sum_{p=1}^{k}a_{pi}\right)\left(\frac{1}{k}\sum_{p=1}^{k}\frac{a_{pi}}{\sqrt{(1-\beta_2)a_{pi}^2 + d_i}}\right)$$

We define two vectors $\mathbf{a}_i, \mathbf{h}_i \in \mathbb{R}^k$ s.t $(\mathbf{a}_i)_p = a_{pi}$ and $(\mathbf{h}_i)_p = \frac{1}{\sqrt{(1-\beta_2)a_{pi}^2 + d_i}}$

Substituting this, the above expression can be written as,

$$\mathbb{E}\left[\langle \nabla f(x_t), V_t^{-1/2}g_t\rangle \mid H_t\right] = \frac{1}{k^2}\sum_{i=1}^{d}\left(\mathbf{a}_i^\top \mathbf{1}_k\right)\left(\mathbf{h}_i^\top \mathbf{a}_i\right) = \frac{1}{k^2}\sum_{i=1}^{d}\mathbf{a}_i^\top\left(\mathbf{1}_k\mathbf{h}_i^\top\right)\mathbf{a}_i \qquad (4)$$

Note that with this substitution, the RHS of the claimed lemma becomes,

$$\mu_{\min}\|\nabla f(x_t)\|^2 = \mu_{\min}\sum_{i=1}^{d}\left(\frac{1}{k}\sum_{p=1}^{k}\nabla_p f(\mathbf{x}_t)\right)^2$$

$$= \frac{\mu_{\min}}{k^2}\sum_{i=1}^{d}(\mathbf{a}_i^\top \mathbf{1}_k)^2$$

$$= \frac{\mu_{\min}}{k^2}\sum_{i=1}^{d}\mathbf{a}_i^\top \mathbf{1}_k\mathbf{1}_k^\top \mathbf{a}_i$$

Therefore our claim is proved if we show that for all $i$, $\frac{1}{k^2}\mathbf{a}_i^\top\left(\mathbf{1}_k\mathbf{h}_i^\top\right)\mathbf{a}_i - \frac{\mu_{\min}}{k^2}\mathbf{a}_i^\top \mathbf{1}_k\mathbf{1}_k^\top \mathbf{a}_i \geq 0$. This can be simplified as,

$$\frac{1}{k^2}\mathbf{a}_i^\top\left(\mathbf{1}_k\mathbf{h}_i^\top\right)\mathbf{a}_i - \mu_{\min}\frac{1}{k^2}\mathbf{a}_i^\top \mathbf{1}_k\mathbf{1}_k^\top \mathbf{a}_i = \frac{1}{k^2}\mathbf{a}_i^\top\left(\mathbf{1}_k\left(\mathbf{h}_i - \mu_{\min}\mathbf{1}_k\right)^\top\right)\mathbf{a}_i$$

To further simplify, we define $\mathbf{q}_i \in \mathbb{R}^k$, $(\mathbf{q}_i)_p = (\mathbf{h}_i)_p - \mu_{\min} = \frac{1}{\sqrt{(1-\beta_2)a_{pi}^2 + d_i}} - \mu_{\min}$. We therefore need to show,

$$\frac{1}{k^2}\mathbf{a}_i^\top\left(\mathbf{1}_k\mathbf{q}_i^\top\right)\mathbf{a}_i \geq 0$$

We first bound $d_i$ by recalling the definition of $\sigma_f$ (from which it follows that $a_{pi}^2 \leq \sigma_f^2$),

$$d_i \le (1 - \beta_2) \left[ \xi + \sum_{k=1}^{t-1} \beta_2^{t-k} (\sigma_f^2 + \xi) \right] = (1 - \beta_2) \left[ \xi + \frac{\beta_2 - \beta_2^{t-1}}{1 - \beta_2} (\sigma_f^2 + \xi) \right]$$

$$\le (1 - \beta_2)\xi + (\beta_2 - \beta_2^{t-1})\xi + (\beta_2 - \beta_2^{t-1})\sigma_f^2 = (1 - \beta_2^{t-1})\xi + (\beta_2 - \beta_2^{t-1})\sigma_f^2$$

$$\implies \sqrt{(1 - \beta_2)a_{pi}^2 + d_i} \le \sqrt{(1 - \beta_2)\sigma_f^2 + (1 - \beta_2^{t-1})\xi + (\beta_2 - \beta_2^{t-1})\sigma_f^2} = \sqrt{(1 - \beta_2^{t-1})(\sigma_f^2 + \xi)}$$

$$\implies -\mu_{\min} + \frac{1}{\sqrt{(1 - \beta_2)a_{pi}^2 + d_i}} \ge -\mu_{\min} + \frac{1}{\sqrt{(1 - \beta_2^{t-1})(\sigma_f^2 + \xi)}} = -\frac{1}{\sqrt{\sigma_f^2 + \xi}} + \frac{1}{\sqrt{(1 - \beta_2^{t-1})(\sigma_f^2 + \xi)}}$$

$$\implies -\mu_{\min} + \frac{1}{\sqrt{(1 - \beta_2)a_{pi}^2 + d_i}} \ge 0 \tag{5}$$

The inequality follows since $\beta_2 \in (0, 1]$

Putting this all together, we get,

$$(\mathbf{a}_i^\top \mathbf{1}_k)(\mathbf{q}_i^\top \mathbf{a}_i)$$

$$= \left( \sum_{p=1}^{k} a_{pi} \right) \left( \sum_{p=1}^{k} \left[ -\mu_{\min} a_{pi} + \frac{a_{pi}}{\sqrt{(1 - \beta_2)a_{pi}^2 + d_i}} \right] \right)$$

$$= \sum_{p,q=1}^{k} \left[ -\mu_{\min} a_{pi} a_{qi} + \frac{a_{pi} a_{qi}}{\sqrt{(1 - \beta_2)a_{pi}^2 + d_i}} \right]$$

$$= \sum_{p,q=1}^{k} a_{pi} a_{qi} \left[ -\mu_{\min} + \frac{1}{\sqrt{(1 - \beta_2)a_{pi}^2 + d_i}} \right]$$

Now our assumption that for all $\mathbf{x}$, $\text{sign}(\nabla f_p(\mathbf{x})) = \text{sign}(\nabla f_q(\mathbf{x}))$ for all $p, q \in \{1, \ldots, k\}$ leads to the conclusion that the term $a_{pi} a_{qi} \ge 0$. And we had already shown in equation 5 that $\left[ -\mu_{\min} + \frac{1}{\sqrt{(1-\beta_2)a_{pi}^2 + d_i}} \right] \ge 0$. Thus we have shown that $(\mathbf{a}_i^\top \mathbf{1}_k)(\mathbf{q}_i^\top \mathbf{a}_i) \ge 0$ and this finishes the proof. $\qquad \square$

## A.2 Proving deterministic RMSProp - the version with standard speed (Proof of Theorem 3.2)

*Proof.* By the $L-$smoothness condition and the update rule in Algorithm 2 we have,

$$f(\mathbf{x}_{t+1}) \leq f(\mathbf{x}_t) - \alpha_t \langle \nabla f(\mathbf{x}_t), V_t^{-\frac{1}{2}} \nabla f(\mathbf{x}_t) \rangle + \alpha_t^2 \frac{L}{2} \|V_t^{-\frac{1}{2}} \nabla f(\mathbf{x}_t)\|^2$$

$$\implies f(\mathbf{x}_{t+1}) - f(\mathbf{x}_t) \leq \alpha_t \left( \frac{L\alpha_t}{2} \|V_t^{-\frac{1}{2}} \nabla f(\mathbf{x}_t)\|^2 - \langle \nabla f(\mathbf{x}_t), V_t^{-\frac{1}{2}} \nabla f(\mathbf{x}_t) \rangle \right) \qquad (6)$$

For $0 < \delta_t^2 < \frac{1}{\sqrt{1-\beta_2^t}\sqrt{\sigma_t^2+\xi}}$ we now show a strictly positive lowerbound on the following function,

$$\frac{2}{L} \left( \frac{\langle \nabla f(\mathbf{x}_t), V_t^{-\frac{1}{2}} \nabla f(\mathbf{x}_t) \rangle - \delta_t^2 \|\nabla f(\mathbf{x}_t)\|^2}{\|V_t^{-\frac{1}{2}} \nabla f(\mathbf{x}_t)\|^2} \right) \qquad (7)$$

We define $\sigma_t := \max_{i=1,..,t} \|\nabla f(\mathbf{x}_i)\|$ and we solve the recursion for $\mathbf{v}_t$ as, $\mathbf{v}_t = (1 - \beta_2) \sum_{k=1}^t \beta_2^{t-k}(\mathbf{g}_k^2 + \xi)$. This lets us write the following bounds,

$$\langle \nabla f(\mathbf{x}_t), V_t^{-\frac{1}{2}} \nabla f(\mathbf{x}_t) \rangle \geq \lambda_{min}(V_t^{-\frac{1}{2}}) \|\nabla f(\mathbf{x}_t)\|^2 \geq \frac{\|\nabla f(\mathbf{x}_t)\|^2}{\sqrt{\max_{i=1,..,d}(\mathbf{v}_t)_i}}$$

$$\geq \frac{\|\nabla f(\mathbf{x}_t)\|^2}{\sqrt{\max_{i=1,..,d}((1-\beta_2)\sum_{k=1}^t \beta_2^{t-k}(\mathbf{g}_k^2 + \xi \mathbf{1}_d)_i)}}$$

$$\geq \frac{\|\nabla f(\mathbf{x}_t)\|^2}{\sqrt{1-\beta_2^t}\sqrt{\sigma_t^2 + \xi}} \qquad (8)$$

Now we define, $\epsilon_t := \min_{k=1,..,t,i=1,..,d}(\nabla f(\mathbf{x}_k))_i^2$ and this lets us get the following sequence of inequalities,

$$\|V_t^{-\frac{1}{2}} \nabla f(\mathbf{x}_t)\|^2 \leq \lambda_{max}^2(V_t^{-\frac{1}{2}}) \|\nabla f(\mathbf{x}_t)\|^2 \leq \frac{\|\nabla f(\mathbf{x}_t)\|^2}{(\min_{i=1,..,d}(\sqrt{(\mathbf{v}_t)_i}))^2} \leq \frac{\|\nabla f(\mathbf{x}_t)\|^2}{(1-\beta_2^t)(\xi + \epsilon_t)} \qquad (9)$$

So combining equations 9 and 8 into equation 7 and from the exit line in the loop we are assured that $\|\nabla f(\mathbf{x}_t)\|^2 \neq 0$ and combining these we have,

$$\frac{2}{L} \left( \frac{-\delta_t^2 \|\nabla f(\mathbf{x}_t)\|^2 + \langle \nabla f(\mathbf{x}_t), V_t^{-\frac{1}{2}} \nabla f(\mathbf{x}_t) \rangle}{\|V_t^{-\frac{1}{2}} \nabla f(\mathbf{x}_t)\|^2} \right) \geq \frac{2}{L} \left( \frac{-\delta_t^2 + \frac{1}{\sqrt{1-\beta_2^t}\sqrt{\sigma_t^2+\xi}}}{\frac{1}{(1-\beta_2^t)(\xi+\epsilon_t)}} \right)$$

$$\geq \frac{2(1-\beta_2^t)(\xi+\epsilon_t)}{L} \left( -\delta_t^2 + \frac{1}{\sqrt{1-\beta_2^t}\sqrt{\sigma_t^2+\xi}} \right)$$

Now our definition of $\delta_t^2$ allows us to define a parameter $0 < \beta_t := \frac{1}{\sqrt{1-\beta_2^t}\sqrt{\sigma_t^2+\xi}} - \delta_t^2$ and rewrite the above equation as,

$$\frac{2}{L} \left( \frac{-\delta_t^2 \|\nabla f(\mathbf{x}_t)\|^2 + \langle \nabla f(\mathbf{x}_t), V_t^{-\frac{1}{2}} \nabla f(\mathbf{x}_t) \rangle}{\|V_t^{-\frac{1}{2}} \nabla f(\mathbf{x}_t)\|^2} \right) \geq \frac{2(1-\beta_2^t)(\xi+\epsilon_t)\beta_t}{L} \qquad (10)$$

We can as well satisfy the conditions needed on the variables, $\beta_t$ and $\delta_t$ by choosing,

$$\delta_t^2 = \frac{1}{2} \min_{t=1,\cdots} \frac{1}{\sqrt{1-\beta_2^t}\sqrt{\sigma_t^2+\xi}} = \frac{1}{2} \frac{1}{\sqrt{\sigma^2+\xi}} =: \delta^2$$

and

$$\beta_t = \min_{t=1,\ldots} \frac{1}{\sqrt{1-\beta_2^t}\sqrt{\sigma_t^2 + \xi}} - \delta^2 = \frac{1}{2}\frac{1}{\sqrt{\sigma^2 + \xi}}$$

Then the worst-case lowerbound in equation 10 becomes,

$$\frac{2}{L}\left(\frac{-\delta_t^2\|\nabla f(\mathbf{x}_t)\|^2 + \langle \nabla f(\mathbf{x}_t), V_t^{-\frac{1}{2}}\nabla f(\mathbf{x}_t)\rangle}{\|V_t^{-\frac{1}{2}}\nabla f(\mathbf{x}_t)\|^2}\right) \geq \frac{2(1-\beta_2)\xi}{L} \times \frac{1}{2}\frac{1}{\sqrt{\sigma^2 + \xi}}$$

This now allows us to see that a constant step length $\alpha_t = \alpha > 0$ can be defined as, $\alpha = \frac{(1-\beta_2)\xi}{L\sqrt{\sigma^2+\xi}}$ and this is such that the above equation can be written as, $\frac{L\alpha}{2}\|V_t^{-\frac{1}{2}}\nabla f(\mathbf{x}_t)\|^2 - \langle \nabla f(\mathbf{x}_t), V_t^{-\frac{1}{2}}\nabla f(\mathbf{x}_t)\rangle \leq -\delta^2 \|\nabla f(\mathbf{x}_t)\|^2$ . This when substituted back into equation 6 we have,

$$f(\mathbf{x}_{t+1}) - f(\mathbf{x}_t) \leq -\delta^2\alpha\|\nabla f(\mathbf{x}_t)\|^2 = -\delta^2\alpha\|\nabla f(\mathbf{x}_t)\|^2$$

This gives us,

$$\|\nabla f(\mathbf{x}_t)\|^2 \leq \frac{1}{\delta^2\alpha}[f(\mathbf{x}_t) - f(\mathbf{x}_{t+1})]$$

$$\implies \sum_{t=1}^{T}\|\nabla f(\mathbf{x}_t)\|^2 \leq \frac{1}{\delta^2\alpha}[f(\mathbf{x}_1) - f(\mathbf{x}_*)] \tag{11}$$

$$\implies \min_{t=1,\ldots T}\|\nabla f(\mathbf{x}_t)\|^2 \leq \frac{1}{T\delta^2\alpha}[f(\mathbf{x}_1) - f(\mathbf{x}_*)] \tag{12}$$

Thus for any given $\epsilon > 0$, $T$ satisfying, $\frac{1}{T\delta^2\alpha}[f(\mathbf{x}_1) - f(\mathbf{x}_*)] \leq \epsilon^2$ is a sufficient condition to ensure that the algorithm finds a point $\mathbf{x}_{result} := \arg\min_{t=1,\ldots T}\|\nabla f(\mathbf{x}_t)\|^2$ with $\|\nabla f(\mathbf{x}_{result})\|^2 \leq \epsilon^2$.

Thus we have shown that using a constant step length of $\alpha = \frac{(1-\beta_2)\xi}{L\sqrt{\sigma^2+\xi}}$ deterministic RMSProp can find an $\epsilon-$critical point in $T = \frac{1}{\epsilon^2} \times \frac{f(\mathbf{x}_1)-f(\mathbf{x}_*)}{\delta^2\alpha} = \frac{1}{\epsilon^2} \times \frac{2L(\sigma^2+\xi)(f(\mathbf{x}_1)-f(\mathbf{x}_*))}{(1-\beta_2)\xi}$ steps.

$\square$

### A.3 Proving deterministic RMSProp - the version with no added shift (Proof of Theorem 3.3)

*Proof.* From the $L-$smoothness condition on $f$ we have between consecutive iterates of the above algorithm,

$$f(\mathbf{x}_{t+1}) \le f(\mathbf{x}_t) - \alpha_t \langle \nabla f(\mathbf{x}_t), V_t^{-\frac{1}{2}} \nabla f(\mathbf{x}_t) \rangle + \frac{L}{2} \alpha_t^2 \| V_t^{-\frac{1}{2}} \nabla f(\mathbf{x}_t) \|^2$$

$$\implies \langle \nabla f(\mathbf{x}_t), V_t^{-\frac{1}{2}} \nabla f(\mathbf{x}_t) \rangle \le \frac{1}{\alpha_t} (f(\mathbf{x}_t) - f(\mathbf{x}_{t+1})) + \frac{L\alpha_t}{2} \| V_t^{-\frac{1}{2}} \nabla f(\mathbf{x}_t) \|^2 \tag{13}$$

$$\tag{14}$$

Now the recursion for $\mathbf{v}_t$ can be solved to get, $\mathbf{v}_t = (1 - \beta_2) \sum_{k=1}^t \beta_2^{t-k} \mathbf{g}_k^2$. Then $\left\| V_t^{\frac{1}{2}} \right\| \ge \frac{1}{\max_{i \in \text{Support}(\mathbf{v}_t)} \sqrt{(\mathbf{v}_t)_i}} = \frac{1}{\max_{i \in \text{Support}(\mathbf{v}_t)} \sqrt{(1-\beta_2) \sum_{k=1}^t \beta_2^{t-k} (\mathbf{g}_k^2)_i}} = \frac{1}{\max_{i \in \text{Support}(\mathbf{v}_t)} \sigma \sqrt{(1-\beta_2) \sum_{k=1}^t \beta_2^{t-k}}} = \frac{1}{\sigma \sqrt{(1-\beta_2^t)}}$. Substituting this in a lowerbound on the LHS of equation 13 we get,

$$\frac{1}{\sigma \sqrt{(1-\beta_2^t)}} \| \nabla f(\mathbf{x}_t) \|^2 \le \langle \nabla f(\mathbf{x}_t), V_t^{-\frac{1}{2}} \nabla f(\mathbf{x}_t) \rangle \le \frac{1}{\alpha_t} (f(\mathbf{x}_t) - f(\mathbf{x}_{t+1})) + \frac{L\alpha_t}{2} \| V_t^{-\frac{1}{2}} \nabla f(\mathbf{x}_t) \|^2$$

Summing the above we get,

$$\sum_{t=1}^T \frac{1}{\sigma \sqrt{(1-\beta_2^t)}} \| \nabla f(\mathbf{x}_t) \|^2 \le \sum_{t=1}^T \frac{1}{\alpha_t} (f(\mathbf{x}_t) - f(\mathbf{x}_{t+1})) + \sum_{t=1}^T \frac{L\alpha_t}{2} \| V_t^{-\frac{1}{2}} \nabla f(\mathbf{x}_t) \|^2 \tag{15}$$

Now we substitute $\alpha_t = \frac{\alpha}{\sqrt{t}}$ and invoke the definition of $B_\ell$ and $B_u$ to write the first term on the RHS of equation 15 as,

$$\sum_{t=1}^T \frac{1}{\alpha_t} [f(\mathbf{x}_t) - f(\mathbf{x}_{t+1})] = \frac{f(\mathbf{x}_1)}{\alpha} + \sum_{t=1}^T \left( \frac{f(\mathbf{x}_{t+1})}{\alpha_{t+1}} - \frac{f(\mathbf{x}_{t+1})}{\alpha_t} \right) - \frac{f(x_{T+1})}{\alpha_{T+1}}$$

$$= \frac{f(\mathbf{x}_1)}{\alpha} - \frac{f(x_{T+1})}{\alpha_{T+1}} + \frac{1}{\alpha} \sum_{t=1}^T f(\mathbf{x}_{t+1})(\sqrt{t+1} - \sqrt{t})$$

$$\le \frac{B_u}{\alpha} - \frac{B_\ell \sqrt{T+1}}{\alpha} + \frac{B_u}{\alpha}(\sqrt{T+1} - 1)$$

Now we bound the second term in the RHS of equation 15 as follows. Lets first define a function $P(T)$ as follows, $P(T) = \sum_{t=1}^T \alpha_t \| V_t^{-\frac{1}{2}} \nabla f(\mathbf{x}_t) \|^2$ and that gives us,

$$P(T) - P(T-1) = \alpha_T \sum_{i=1}^d \frac{\mathbf{g}_{T,i}^2}{\mathbf{v}_{T,i}} = \alpha_T \sum_{i=1}^d \frac{\mathbf{g}_{T,i}^2}{(1-\beta_2) \sum_{k=1}^T \beta_2^{T-k} \mathbf{g}_{k,i}^2}$$

$$= \frac{\alpha_T}{(1-\beta_2)} \sum_{i=1}^d \frac{\mathbf{g}_{T,i}^2}{\sum_{k=1}^T \beta_2^{T-k} \mathbf{g}_{k,i}^2} \le \frac{d\alpha}{(1-\beta_2)\sqrt{T}}$$

$$\implies \sum_{t=2}^T [P(t) - P(t-1)] = P(T) - P(1) \le \frac{d\alpha}{(1-\beta_2)} \sum_{t=2}^T \frac{1}{\sqrt{t}} \le \frac{d\alpha}{2(1-\beta_2)}(\sqrt{T} - 2)$$

$$\implies P(T) \le P(1) + \frac{d\alpha}{2(1-\beta_2)}(\sqrt{T} - 2)$$

So substituting the above two bounds back into the RHS of the above inequality 15and removing the factor of $\sqrt{1 - \beta_2^T} < 1$ from the numerator, we can define a point $\mathbf{x}_{result}$ as follows,

$$\|\nabla f(\mathbf{x}_{result})\|^2 := \underset{t=1,..,T}{\arg\min} \|\nabla f(\mathbf{x}_t)\|^2 \leq \frac{1}{T} \sum_{t=1}^{T} \|\nabla f(\mathbf{x}_t)\|^2$$
$$\leq \frac{\sigma}{T} \left( \frac{B_u}{\alpha} - \frac{B_l \sqrt{T+1}}{\alpha} + \frac{B_u}{\alpha}(\sqrt{T+1} - 1) \right.$$
$$\left. + \frac{L}{2} \left[ P(1) + \frac{d\alpha}{2(1-\beta_2)}(\sqrt{T} - 2) \right] \right)$$

Thus it follows that for $T = O(\frac{1}{\epsilon^4})$ the algorithm 2 is guaranteed to have found at least one point $\mathbf{x}_{result}$ such that, $\|\nabla f(\mathbf{x}_{result})\|^2 \leq \epsilon^2$ $\qquad\square$

## A.4 PROVING ADAM (PROOF OF THEOREM 3.4)

*Proof.*

Let us assume to the contrary that $\|g_t\| > \epsilon$ for all $t = 1, 2, 3. \ldots$. We will show that this assumption will lead to a contradiction. By $L-$smoothness of the objective we have the following relationship between the values at consecutive updates,

$$f(\mathbf{x}_{t+1}) \leq f(\mathbf{x}_t) + \langle \nabla f(\mathbf{x}_t), \mathbf{x}_{t+1} - \mathbf{x}_t \rangle + \frac{L}{2} \|\mathbf{x}_{t+1} - \mathbf{x}_t\|^2$$

Substituting the update rule using a dummy step length $\eta_t > 0$ we have,

$$f(\mathbf{x}_{t+1}) \leq f(\mathbf{x}_t) - \eta_t \langle \nabla f(\mathbf{x}_t), \left(V_t^{\frac{1}{2}} + \text{diag}(\xi \mathbf{1}_d)\right)^{-1} \mathbf{m}_t \rangle + \frac{L\eta_t^2}{2} \left\| \left(V_t^{\frac{1}{2}} + \text{diag}(\xi \mathbf{1}_d)\right)^{-1} \mathbf{m}_t \right\|^2$$
$$\implies f(\mathbf{x}_{t+1}) - f(\mathbf{x}_t) \leq \eta_t \left( -\langle \mathbf{g}_t, \left(V_t^{\frac{1}{2}} + \text{diag}(\xi \mathbf{1}_d)\right)^{-1} \mathbf{m}_t \rangle + \frac{L\eta_t}{2} \left\| \left(V_t^{\frac{1}{2}} + \text{diag}(\xi \mathbf{1}_d)\right)^{-1} \mathbf{m}_t \right\|^2 \right)$$
$$(16)$$

The RHS in equation 16 above is a quadratic in $\eta_t$ with two roots: $0$ and $\dfrac{\langle \mathbf{g}_t, \left(V_t^{\frac{1}{2}} + \text{diag}(\xi \mathbf{1}_d)\right)^{-1} \mathbf{m}_t \rangle}{\frac{L}{2} \left\| \left(V_t^{\frac{1}{2}} + \text{diag}(\xi \mathbf{1}_d)\right)^{-1} \mathbf{m}_t \right\|^2}$.

So the quadratic's minimum value is at the midpoint of this interval, which gives us a candidate $t^{th}-$step length i.e $\alpha_t^* := \frac{1}{2} \cdot \dfrac{\langle \mathbf{g}_t, \left(V_t^{\frac{1}{2}} + \text{diag}(\xi \mathbf{1}_d)\right)^{-1} \mathbf{m}_t \rangle}{\frac{L}{2} \left\| \left(V_t^{\frac{1}{2}} + \text{diag}(\xi \mathbf{1}_d)\right)^{-1} \mathbf{m}_t \right\|^2}$ and the value of the quadratic at this point is $-\frac{1}{4} \cdot \dfrac{(\langle \mathbf{g}_t, \left(V_t^{\frac{1}{2}} + \text{diag}(\xi \mathbf{1}_d)\right)^{-1} \mathbf{m}_t \rangle)^2}{\frac{L}{2} \left\| \left(V_t^{\frac{1}{2}} + \text{diag}(\xi \mathbf{1}_d)\right)^{-1} \mathbf{m}_t \right\|^2}$. That is with step lengths being this $\alpha_t^*$ we have the following guarantee of decrease of function value between consecutive steps,

$$f(\mathbf{x}_{t+1}) - f(\mathbf{x}_t) \leq -\frac{1}{2L} \cdot \frac{(\langle \mathbf{g}_t, \left(V_t^{\frac{1}{2}} + \text{diag}(\xi \mathbf{1}_d)\right)^{-1} \mathbf{m}_t \rangle)^2}{\left\| \left(V_t^{\frac{1}{2}} + \text{diag}(\xi \mathbf{1}_d)\right)^{-1} \mathbf{m}_t \right\|^2} \qquad (17)$$

Now we separately lower bound the numerator and upper bound the denominator of the RHS above.

**Upperbound on** $\left\| \left( V_t^{\frac{1}{2}} + \mathbf{diag}(\xi \mathbf{1}_d) \right)^{-1} \mathbf{m}_t \right\|$

We have, $\lambda_{max}\left( \left( V_t^{\frac{1}{2}} + \text{diag}(\xi \mathbf{1}_d) \right)^{-1} \right) \leq \frac{1}{\xi + \min_{i=1..d} \sqrt{(\mathbf{v}_t)_i}}$ Further we note that the recursion of $\mathbf{v}_t$ can be solved as, $\mathbf{v}_t = (1 - \beta_2) \sum_{k=1}^{t} \beta_2^{t-k} \mathbf{g}_k^2$. Now we define, $\epsilon_t := \min_{k=1,..,t, i=1,..,d}(\mathbf{g}_k^2)_i$ and this gives us,

$$\lambda_{max}\left( \left( V_t^{\frac{1}{2}} + \text{diag}(\xi \mathbf{1}_d) \right)^{-1} \right) \leq \frac{1}{\xi + \sqrt{(1 - \beta_2^t)\epsilon_t}} \tag{18}$$

We solve the recursion for $\mathbf{m}_t$ to get, $\mathbf{m}_t = (1 - \beta_1) \sum_{k=1}^{t} \beta_1^{t-k} \mathbf{g}_k$. Then by triangle inequality and defining $\sigma_t := \max_{i=1,..,t} \|\nabla f(\mathbf{x}_i)\|$ we have, $\|\mathbf{m}_t\| \leq (1 - \beta_1^t)\sigma_t$. Thus combining this estimate of $\|\mathbf{m}_t\|$ with equation 18 we have,

$$\left\| \left( V_t^{\frac{1}{2}} + \text{diag}(\xi \mathbf{1}_d) \right)^{-1} \mathbf{m}_t \right\| \leq \frac{(1 - \beta_1^t)\sigma_t}{\xi + \sqrt{\epsilon_t(1 - \beta_2^t)}} \leq \frac{(1 - \beta_1^t)\sigma_t}{\xi} \tag{19}$$

**Lowerbound on** $\langle \mathbf{g}_t, \left( V_t^{\frac{1}{2}} + \mathbf{diag}(\xi \mathbf{1}_d) \right)^{-1} \mathbf{m}_t \rangle$

To analyze this we define the following sequence of functions for each $i = 0, 1, 2.., t$

$$Q_i = \langle \mathbf{g}_t, \left( V_t^{\frac{1}{2}} + \text{diag}(\xi \mathbf{1}_d) \right)^{-1} \mathbf{m}_i \rangle$$

This gives us the following on substituting the update rule for $\mathbf{m}_t$,

$$Q_i - \beta_1 Q_{i-1} = \langle \mathbf{g}_t, \left( V_t^{\frac{1}{2}} + \text{diag}(\xi \mathbf{1}_d) \right)^{-1} (\mathbf{m}_i - \beta_1 \mathbf{m}_{i-1}) \rangle$$
$$= (1 - \beta_1)\langle \mathbf{g}_t, \left( V_t^{\frac{1}{2}} + \text{diag}(\xi \mathbf{1}_d) \right)^{-1} \mathbf{g}_i \rangle$$

At $i = t$ we have,

$$Q_t - \beta_1 Q_{t-1} \geq (1 - \beta_1) \|\mathbf{g}_t\|^2 \lambda_{min}\left( \left( V_t^{\frac{1}{2}} + \text{diag}(\xi \mathbf{1}_d) \right)^{-1} \right)$$

Lets define, $\sigma_{t-1} := \max_{i=1,..,t-1} \|\nabla f(\mathbf{x}_i)\|$ and this gives us for $i \in \{1, .., t-1\}$,

$$Q_i - \beta_1 Q_{i-1} \geq -(1 - \beta_1) \|\mathbf{g}_t\| \sigma_{t-1} \lambda_{max}\left( \left( V_t^{\frac{1}{2}} + \text{diag}(\xi \mathbf{1}_d) \right)^{-1} \right)$$

We note the following identity,

$$Q_t - \beta_1^t Q_0 = (Q_t - \beta_1 Q_{t-1}) + \beta_1 (Q_{t-1} - \beta_1 Q_{t-2}) + \beta_1^2 (Q_{t-2} - \beta_1 Q_{t-3}) + .. + \beta_1^{t-1}(Q_1 - \beta_1 Q_0)$$

Now we use the lowerbounds proven on $Q_i - \beta_1 Q_{i-1}$ for $i \in \{1, .., t-1\}$ and $Q_t - \beta_1 Q_{t-1}$ to lowerbound the above sum as,

$$Q_t - \beta_1^t Q_0 \geq (1 - \beta_1) \|\mathbf{g}_t\|^2 \lambda_{min}\left(\left(V_t^{\frac{1}{2}} + \text{diag}(\xi \mathbf{1}_d)\right)^{-1}\right)$$

$$- (1 - \beta_1) \|\mathbf{g}_t\| \sigma_{t-1} \lambda_{max}\left(\left(V_t^{\frac{1}{2}} + \text{diag}(\xi \mathbf{1}_d)\right)^{-1}\right) \sum_{j=1}^{t-1} \beta_1^j$$

$$\geq (1 - \beta_1) \|\mathbf{g}_t\|^2 \lambda_{min}\left(\left(V_t^{\frac{1}{2}} + \text{diag}(\xi \mathbf{1}_d)\right)^{-1}\right)$$

$$- (\beta_1 - \beta_1^t) \|\mathbf{g}_t\| \sigma_{t-1} \lambda_{max}\left(\left(V_t^{\frac{1}{2}} + \text{diag}(\xi \mathbf{1}_d)\right)^{-1}\right) \tag{20}$$

We can evaluate the following lowerbound, $\lambda_{min}\left(\left(V_t^{\frac{1}{2}} + \text{diag}(\xi \mathbf{1}_d)\right)^{-1}\right) \geq \frac{1}{\xi + \sqrt{\max_{i=1,..,d}(\mathbf{v}_t)_i}}$.
Next we remember that the recursion of $\mathbf{v}_t$ can be solved as, $\mathbf{v}_t = (1 - \beta_2) \sum_{k=1}^{t} \beta_2^{t-k} \mathbf{g}_k^2$ and we define, $\sigma_t := \max_{i=1,..,t} \|\nabla f(\mathbf{x}_i)\|$ to get,

$$\lambda_{min}\left(\left(V_t^{\frac{1}{2}} + \text{diag}(\xi \mathbf{1}_d)\right)^{-1}\right) \geq \frac{1}{\xi + \sqrt{(1 - \beta_2^t)\sigma_t^2}} \tag{21}$$

Now we combine the above and equation 18 and the known value of $Q_0 = 0$ (from definition and initial conditions) to get from the equation 20,

$$Q_t \geq -(\beta_1 - \beta_1^t) \|\mathbf{g}_t\| \sigma_{t-1} \frac{1}{\xi + \sqrt{(1 - \beta_2^t)\epsilon_t}}$$

$$+ (1 - \beta_1) \|\mathbf{g}_t\|^2 \frac{1}{\xi + \sqrt{(1 - \beta_2^t)\sigma_t^2}}$$

$$\geq \|\mathbf{g}_t\|^2 \left(\frac{(1 - \beta_1)}{\xi + \sigma\sqrt{(1 - \beta_2^t)}} - \frac{(\beta_1 - \beta_1^t)\sigma}{\xi \|\mathbf{g}_t\|}\right) \tag{22}$$

In the above inequalities we have set $\epsilon_t = 0$ and we have set, $\sigma_t = \sigma_{t-1} = \sigma$. Now we examine the following part of the lowerbound proven above,

$$\frac{(1 - \beta_1)}{\xi + \sqrt{(1 - \beta_2^t)\sigma^2}} - \frac{(\beta_1 - \beta_1^t)\sigma}{\xi \|\mathbf{g}_t\|} = \frac{\xi \|\mathbf{g}_t\| (1 - \beta_1) - \sigma(\beta_1 - \beta_1^t)(\xi + \sigma\sqrt{(1 - \beta_2^t)})}{\xi \|\mathbf{g}_t\| (\xi + \sigma\sqrt{(1 - \beta_2^t)})}$$

$$= \sigma(\beta_1 - \beta_1^t) \frac{\xi\left(\frac{\|\mathbf{g}_t\|(1 - \beta_1)}{\sigma(\beta_1 - \beta_1^t)} - 1\right) - \sigma\sqrt{(1 - \beta_2^t)}}{\xi \|\mathbf{g}_t\| (\xi + \sigma\sqrt{(1 - \beta_2^t)})}$$

$$= \sigma(\beta_1 - \beta_1^t)\left(\frac{\|\mathbf{g}_t\|(1 - \beta_1)}{\sigma(\beta_1 - \beta_1^t)} - 1\right) \frac{\xi - \left(\frac{\sigma\sqrt{(1 - \beta_2^t)}}{-1 + \frac{(1 - \beta_1)\|\mathbf{g}_t\|}{(\beta_1 - \beta_1^t)\sigma}}\right)}{\xi \|\mathbf{g}_t\| (\xi + \sigma\sqrt{(1 - \beta_2^t)})}$$

Now we remember the assumption that we are working under i.e $\|\mathbf{g}_t\| > \epsilon$. Also by definition $0 < \beta_1 < 1$ and hence we have $0 < \beta_1 - \beta_1^t < \beta_1$. This implies, $\frac{(1 - \beta_1)\|\mathbf{g}_t\|}{(\beta_1 - \beta_1^t)\sigma} > \frac{(1 - \beta_1)\epsilon}{\beta_1\sigma} > 1$ where the last inequality follows because of our choice of $\epsilon$ as stated in the theorem statement. This allows us to define a constant, $\frac{\epsilon(1 - \beta_1)}{\beta_1\sigma} - 1 := \theta_1 > 0$ s.t $\frac{(1 - \beta_1)\|\mathbf{g}_t\|}{(\beta_1 - \beta_1^t)\sigma} - 1 > \theta_1$

Similarly our definition of $\xi$ allows us to define a constant $\theta_2 > 0$ to get,

$$\left(\frac{\sigma\sqrt{(1 - \beta_2^t)}}{-1 + \frac{(1 - \beta_1)\|\mathbf{g}_t\|}{(\beta_1 - \beta_1^t)\sigma}}\right) < \frac{\sigma}{\theta_1} = \xi - \theta_2$$

Putting the above back into the lowerbound for $Q_t$ in equation 22 we have,

$$Q_t \geq \|\mathbf{g}_t\|^2 \left( \frac{\sigma(\beta_1 - \beta_1^2)\theta_1\theta_2}{\xi\sigma(\xi + \sigma)} \right) \tag{23}$$

Now we substitute the above and equation 19 into equation 17 to get,

$$f(\mathbf{x}_{t+1}) - f(\mathbf{x}_t) \leq -\frac{1}{2L} \cdot \frac{(\langle \mathbf{g}_t, \left( V_t^{\frac{1}{2}} + \text{diag}(\xi \mathbf{1}_d) \right)^{-1} \mathbf{m}_t \rangle)^2}{\left\| \left( V_t^{\frac{1}{2}} + \text{diag}(\xi \mathbf{1}_d) \right)^{-1} \mathbf{m}_t \right\|^2}$$

$$\leq -\frac{1}{2L} \frac{Q_t^2}{\left\| \left( V_t^{\frac{1}{2}} + \text{diag}(\xi \mathbf{1}_d) \right)^{-1} \mathbf{m}_t \right\|^2}$$

$$\leq -\frac{1}{2L} \frac{\|\mathbf{g}_t\|^4 \left( \frac{(\beta_1 - \beta_1^2)\theta_1\theta_2}{\xi(\xi+\sigma)} \right)^2}{\left( \frac{(1-\beta_1^t)\sigma}{\xi} \right)^2}$$

$$\leq -\frac{\|\mathbf{g}_t\|^4}{2L} \left( \frac{(\beta_1 - \beta_1^2)^2\theta_1^2\theta_2^2}{(\xi + \sigma)^2(1 - \beta_1^t)^2\sigma^2} \right) \tag{24}$$

$$\implies \left( \frac{(\beta_1 - \beta_1^2)^2\theta_1^2\theta_2^2}{2L(\xi + \sigma)^2(1 - \beta_1^t)^2\sigma^2} \right) \|\nabla f(\mathbf{x}_t)\|^4 \leq [f(\mathbf{x}_t) - f(\mathbf{x}_{t+1})]$$

$$\implies \sum_{t=2}^{T} \left( \frac{(\beta_1 - \beta_1^2)^2\theta_1^2\theta_2^2}{2L(\xi + \sigma)^2\sigma^2} \right) \|\nabla f(\mathbf{x}_t)\|^4 \leq [f(\mathbf{x}_2) - f(\mathbf{x}_{T+1})]$$

$$\implies \min_{t=2,..,T} \|\nabla f(\mathbf{x}_t)\|^4 \leq \frac{2L(\xi + \sigma)^2\sigma^2}{T(\beta_1 - \beta_1^2)^2\theta_1^2\theta_2^2} [f(\mathbf{x}_2) - f(\mathbf{x}_*)]$$

Observe that if,

$$T \geq \frac{2L\sigma^2(\xi + \sigma)^2}{2\epsilon^4(\beta_1 - \beta_1^2)^2\theta_1^2\theta_2^2} [f(\mathbf{x}_2) - f(\mathbf{x}_*)]$$

then the RHS of the inequality above is less than or equal to $\epsilon^4$ and this would contradict the assumption that $\|\nabla f(\mathbf{x}_t)\| > \epsilon$ for all $t = 1, 2, \ldots$.

As a consequence we have proven the first part of the theorem which guarantees the existence of positive step lengths, $\alpha_t$ s.t ADAM finds an approximately critical point in finite time.

Now choose $\theta_1 = 1$ i.e $\frac{\epsilon}{2} = \frac{\beta_1\sigma}{1-\beta_1}$ i.e $\beta_1 = \frac{\epsilon}{\epsilon+2\sigma}$ $\implies \beta_1(1 - \beta_1) = \frac{\epsilon}{\epsilon+2\sigma}(1 - \frac{\epsilon}{\epsilon+2\sigma}) = \frac{2\sigma\epsilon}{(\epsilon+2\sigma)^2}$. This also gives a easier-to-read condition on $\xi$ in terms of these parameters i.e $\xi > \sigma$. Now choose $\xi = 2\sigma$ i.e $\theta_2 = \sigma$ and making these substitutions gives us,

$$T \geq \frac{18L\sigma^4}{2\epsilon^4 \left( \frac{2\sigma\epsilon}{(\epsilon+2\sigma)} \right)^2 \sigma^2} [f(\mathbf{x}_2) - f(\mathbf{x}_*)] \geq \frac{18L}{8\epsilon^6 \left( \frac{1}{\epsilon+2\sigma} \right)^2} [f(\mathbf{x}_2) - f(\mathbf{x}_*)] \geq \frac{9L\sigma^2}{\epsilon^6} [f(\mathbf{x}_2) - f(\mathbf{x}_*)]$$

We substitute these choices in the step length found earlier to get,

$$\alpha_t^* = \frac{1}{L} \cdot \frac{\langle \mathbf{g}_t, \left(V_t^{\frac{1}{2}} + \text{diag}(\xi \mathbf{1}_d)\right)^{-1} \mathbf{m}_t \rangle}{\left\| \left(V_t^{\frac{1}{2}} + \text{diag}(\xi \mathbf{1}_d)\right)^{-1} \mathbf{m}_t \right\|^2} = \frac{1}{L} \cdot \frac{Q_t}{\left\| \left(V_t^{\frac{1}{2}} + \text{diag}(\xi \mathbf{1}_d)\right)^{-1} \mathbf{m}_t \right\|^2}$$

$$\geq \frac{1}{L} \frac{\|\mathbf{g}_t\|^2 \left( \frac{\sigma^2(\beta_1 - \beta_1^2)}{\xi \sigma (\xi + \sigma)} \right)}{\left( \frac{(1 - \beta_1^t)\sigma}{\xi} \right)^2} = \frac{\|\mathbf{g}_t\|^2}{L(1 - \beta_1^t)^2} \frac{4\epsilon}{3(\epsilon + 2\sigma)^2} := \alpha_t$$

In the theorem statement we choose to call as the final $\alpha_t$ the lowerbound proven above. We check below that this smaller value of $\alpha_t$ still guarantees a decrease in the function value that is sufficient for the statement of the theorem to hold.

**A consistency check!** Let us substitute the above final value of the step length $\alpha_t = \frac{1}{L} \frac{\|\mathbf{g}_t\|^2 \left( \frac{\sigma^2(\beta_1 - \beta_1^2)}{\xi \sigma (\xi + \sigma)} \right)}{\left( \frac{(1 - \beta_1^t)\sigma}{\xi} \right)^2} = \frac{\xi}{L(1 - \beta_1^t)^2} \|\mathbf{g}_t\|^2 \left( \frac{(\beta_1 - \beta_1^2)}{\sigma(\xi + \sigma)} \right)$, the bound in equation 19 (with $\sigma_t$ replaced by $\sigma$), and the bound in equation 23 (at the chosen values of $\theta_1 = 1$ and $\theta_2 = \sigma$) in the original equation 17 to measure the decrease in the function value between consecutive steps,

$$f(\mathbf{x}_{t+1}) - f(\mathbf{x}_t) \leq \alpha_t \left( -\langle \mathbf{g}_t, \left(V_t^{\frac{1}{2}} + \text{diag}(\xi \mathbf{1}_d)\right)^{-1} \mathbf{m}_t \rangle + \frac{L\alpha_t}{2} \left\| \left(V_t^{\frac{1}{2}} + \text{diag}(\xi \mathbf{1}_d)\right)^{-1} \mathbf{m}_t \right\|^2 \right)$$

$$\leq \alpha_t \left( -Q_t + \frac{L\alpha_t}{2} \left\| \left(V_t^{\frac{1}{2}} + \text{diag}(\xi \mathbf{1}_d)\right)^{-1} \mathbf{m}_t \right\|^2 \right)$$

$$\leq \frac{\xi}{L(1 - \beta_1^t)^2} \|\mathbf{g}_t\|^2 \left( \frac{(\beta_1 - \beta_1^2)}{\sigma(\xi + \sigma)} \right) \left( -\|\mathbf{g}_t\|^2 \left( \frac{\sigma(\beta_1 - \beta_1^2)\theta_1 \theta_2}{\xi \sigma(\xi + \sigma)} \right) \right)$$

$$+ \frac{L}{2} \left( \frac{\xi}{L(1 - \beta_1^t)^2} \|\mathbf{g}_t\|^2 \left( \frac{(\beta_1 - \beta_1^2)}{\sigma(\xi + \sigma)} \right) \frac{(1 - \beta_1^t)\sigma}{\xi} \right)^2$$

The RHS above can be simplified to be shown to be equal to the RHS in equation 24 at the same values of $\theta_1$ and $\theta_2$ as used above. And we remember that the bound on the running time was derived from this equation 24. □

## B    HYPERPARAMETER TUNING

Here we describe how we tune the hyper-parameters of each optimization algorithm. NAG has two hyper-parameters, the step size $\alpha$ and the momentum $\mu$. The main hyper-parameters for RMSProp are the step size $\alpha$, the decay parameter $\beta_2$ and the perturbation $\xi$. ADAM, in addition to the ones in RMSProp, also has a momentum parameter $\beta_1$. We vary the step-sizes of ADAM in the conventional way of $\alpha_t = \alpha\sqrt{1 - \beta_2^t}/(1 - \beta_1^t)$.

For tuning the step size, we follow the same method used in Wilson et al. (2017). We start out with a logarithmically-spaced grid of five step sizes. If the best performing parameter was at one of the extremes of the grid, we tried new grid points so that the best performing parameters were at one of the middle points in the grid. While it is computationally infeasible even with substantial resources to follow a similarly rigorous tuning process for all other hyper-parameters, we do tune over them somewhat as described below.

**NAG**    The initial set of step sizes used for NAG were: $\{3e-3, 1e-3, 3e-4, 1e-4, 3e-5\}$. We tune the momentum parameter over values $\mu \in \{0.9, 0.99\}$.

**RMSProp**    The initial set of step sizes used were: $\{3e-4, 1e-4, 3e-5, 1e-5, 3e-6\}$. We tune over $\beta_2 \in \{0.9, 0.99\}$. We set the perturbation value $\xi = 10^{-10}$, following the default values in TensorFlow, except for the experiments in Section 5.1. In Section 5.1, we show the effect on convergence and generalization properties of ADAM and RMSProp when changing this parameter $\xi$.

Note that ADAM and RMSProp uses an accumulator for keeping track of decayed squared gradient $\mathbf{v}_t$. For ADAM this is recommended to be initialized at $\mathbf{v}_0 = 0$. However, we found in the TensorFlow implementation of RMSProp that it sets $\mathbf{v}_0 = \mathbf{1}_d$. Instead of using this version of the algorithm, we used a modified version where we set $\mathbf{v}_0 = 0$. We typically found setting $\mathbf{v}_0 = 0$ to lead to faster convergence in our experiments.

**ADAM**    The initial set of step sizes used were: $\{3e-4, 1e-4, 3e-5, 1e-5, 3e-6\}$. For ADAM, we tune over $\beta_1$ values of $\{0.9, 0.99\}$. For ADAM, We set $\beta_2 = 0.999$ for all our experiments as is set as the default in TensorFlow. Unless otherwise specified we use for the perturbation value $\xi = 10^{-8}$ for ADAM, following the default values in TensorFlow.

Contrary to what is the often used values of $\beta_1$ for ADAM (usually set to 0.9), we found that we often got better results on the autoencoder problem when setting $\beta_1 = 0.99$.

## C    EFFECT OF THE $\xi$ PARAMETER ON ADAPTIVE GRADIENT ALGORITHMS

In Figure 5, we show the same effect of changing $\xi$ as in Section 5.1 on a 1 hidden layer network of 1000 nodes, while keeping all other hyper-parameters fixed (such as learning rate, $\beta_1$, $\beta_2$). These other hyper-parameter values were fixed at the best values of these parameters for the default values of $\xi$, i.e., $\xi = 10^{-10}$ for RMSProp and $\xi = 10^{-8}$ for ADAM.

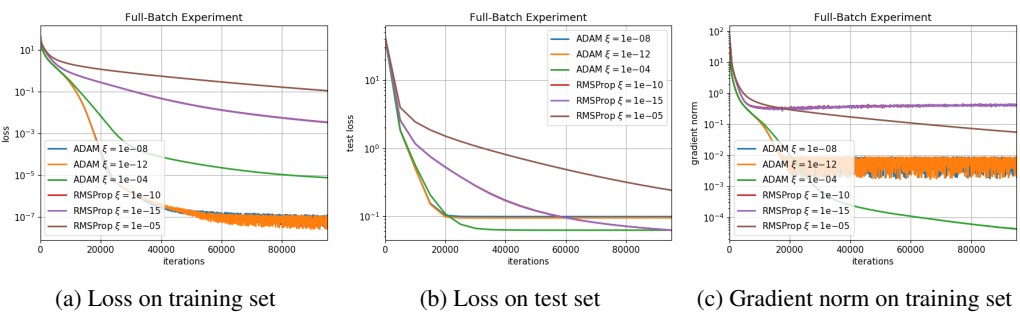

(a) Loss on training set        (b) Loss on test set        (c) Gradient norm on training set

Figure 5: Fixed parameters with changing $\xi$ values. 1 hidden layer network of 1000 nodes

# D ADDITIONAL EXPERIMENTS

## D.1 ADDITIONAL FULL-BATCH EXPERIMENTS ON $22 \times 22$ SIZED IMAGES

In Figures 6, 7 and 8, we show training loss, test loss and gradient norm results for a variety of additional network architectures. Across almost all network architectures, our main results remain consistent. ADAM with $\beta_1 = 0.99$ consistently reaches lower training loss values as well as better generalization than NAG.

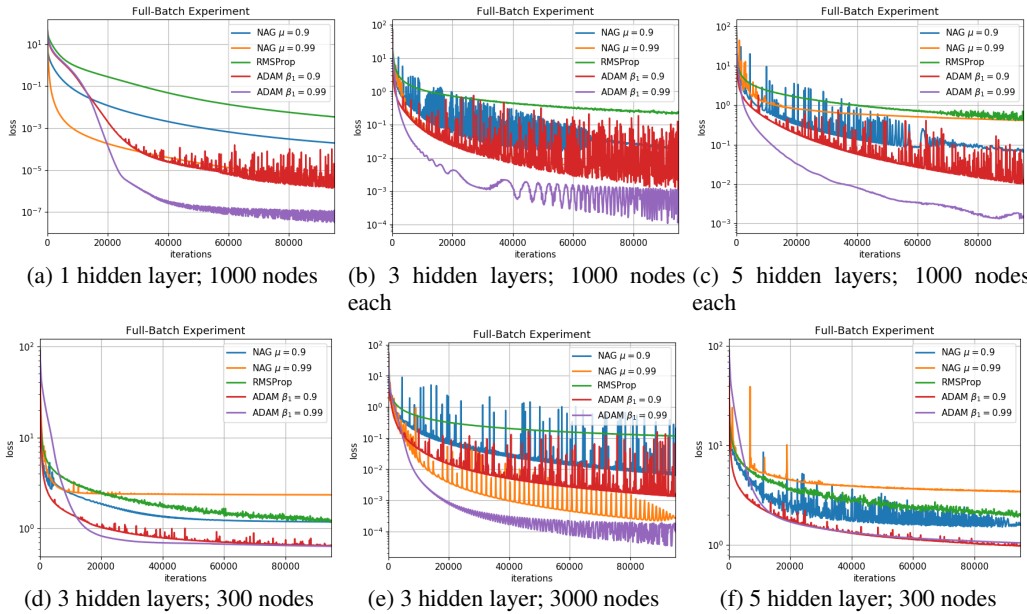

(a) 1 hidden layer; 1000 nodes  (b) 3 hidden layers; 1000 nodes each  (c) 5 hidden layers; 1000 nodes each

(d) 3 hidden layers; 300 nodes  (e) 3 hidden layer; 3000 nodes  (f) 5 hidden layer; 300 nodes

Figure 6: Loss on training set; Input image size $22 \times 22$

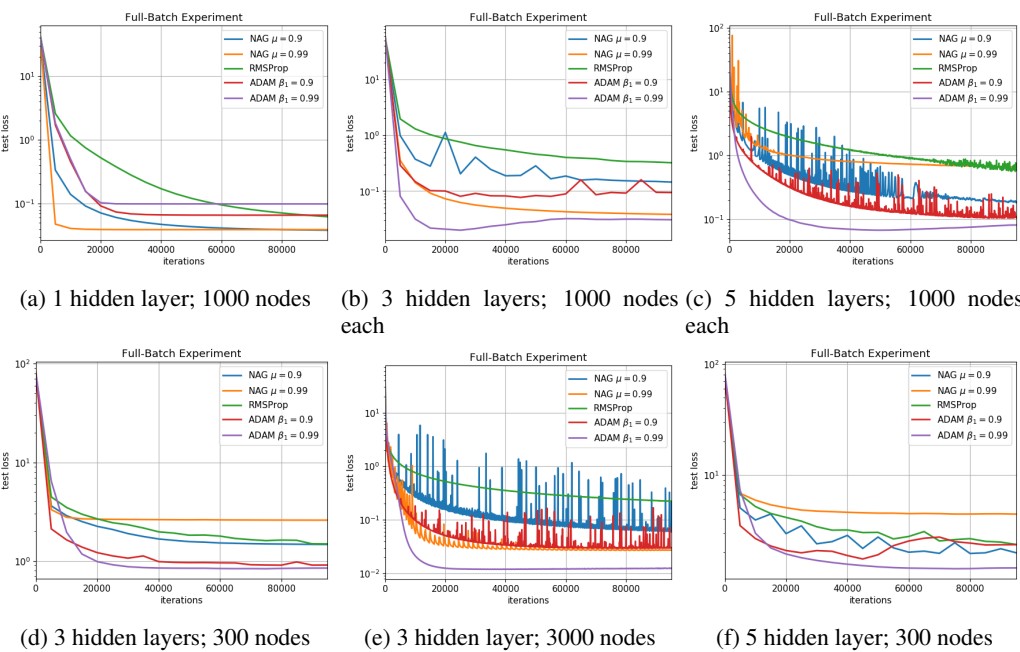

(a) 1 hidden layer; 1000 nodes  (b) 3 hidden layers; 1000 nodes each  (c) 5 hidden layers; 1000 nodes each

(d) 3 hidden layers; 300 nodes  (e) 3 hidden layer; 3000 nodes  (f) 5 hidden layer; 300 nodes

Figure 7: Loss on test set; Input image size $22 \times 22$

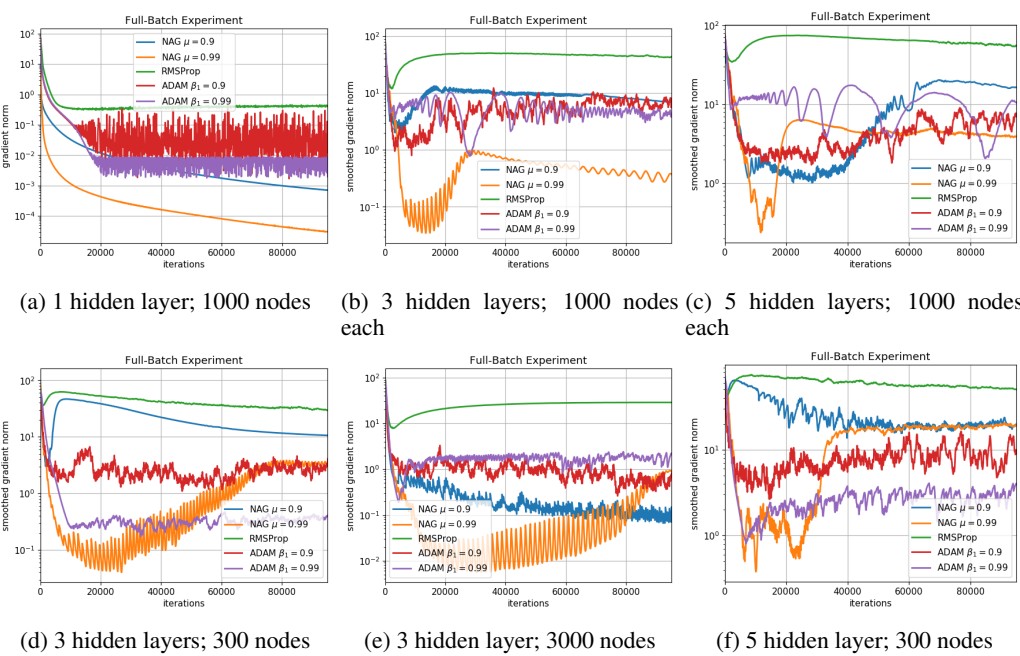

Figure 8: Norm of gradient on training set; Input image size $22 \times 22$

## D.2 ARE THE FULL-BATCH RESULTS CONSISTENT ACROSS DIFFERENT INPUT DIMENSIONS?

To test whether our conclusions are consistent across different input dimensions, we do two experiments where we resize the $22 \times 22$ MNIST image to $17 \times 17$ and to $12 \times 12$. Resizing is done using TensorFlow's `tf.image.resize_images` method, which uses bilinear interpolation.

### D.2.1 INPUT IMAGES OF SIZE $17 \times 17$

Figure 9 shows results on input images of size $17 \times 17$ on a 3 layer network with 1000 hidden nodes in each layer. Our main results extend to this input dimension, where we see ADAM with $\beta_1 = 0.99$ both converging the fastest as well as generalizing the best, while NAG does better than ADAM with $\beta_1 = 0.9$.

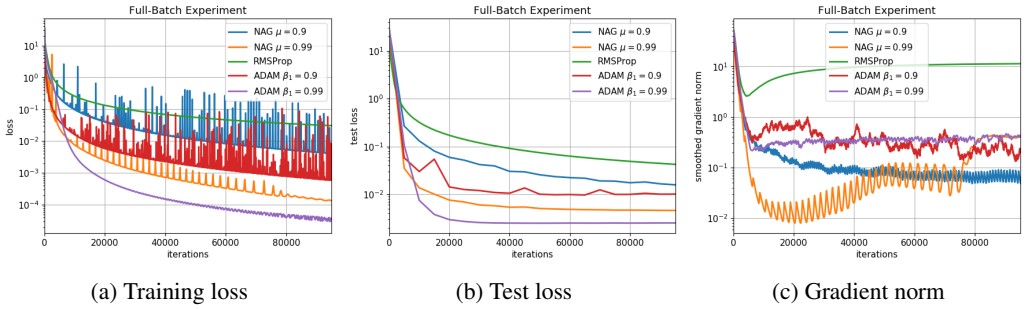

| (a) Training loss | (b) Test loss | (c) Gradient norm |

Figure 9: Full-batch experiments with input image size $17 \times 17$

### D.2.2 INPUT IMAGES OF SIZE $12 \times 12$

Figure 10 shows results on input images of size $12 \times 12$ on a 3 layer network with 1000 hidden nodes in each layer. Our main results extend to this input dimension as well. ADAM with $\beta_1 = 0.99$ converges the fastest as well as generalizes the best, while NAG does better than ADAM with $\beta_1 = 0.9$.

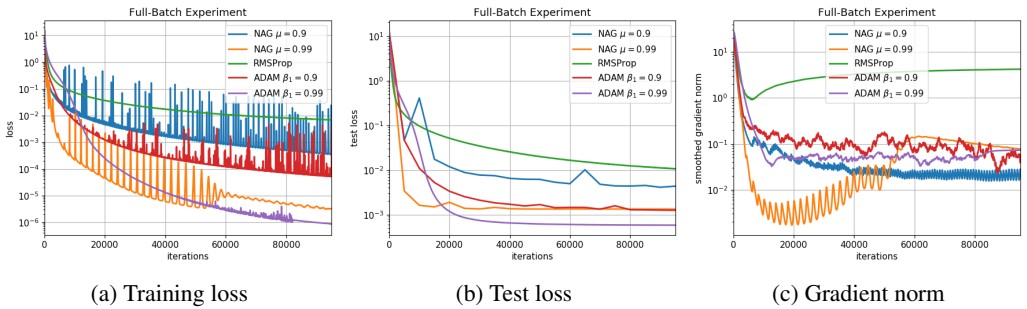

| (a) Training loss | (b) Test loss | (c) Gradient norm |

Figure 10: Full-batch experiments with input image size $12 \times 12$

### D.3 ADDITIONAL MINI-BATCH EXPERIMENTS ON $22 \times 22$ SIZED IMAGES

In Figure 11, we present results on additional neural net architectures on mini-batches of size 100 with an input dimension of $22 \times 22$. We see that most of our full-batch results extend to the mini-batch case.

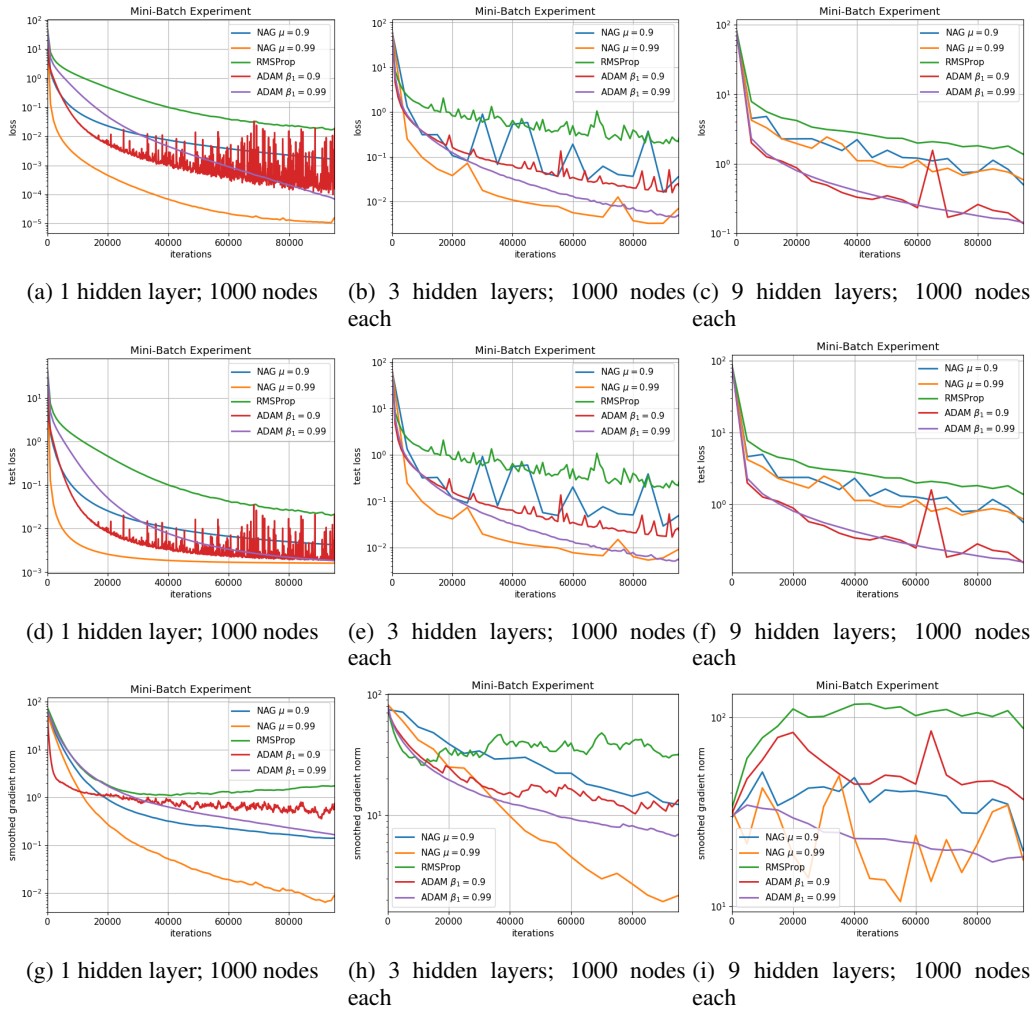

Figure 11: Experiments on various networks with mini-batch size 100 on full MNIST dataset with input image size $22 \times 22$. First row shows the loss on the full training set, middle row shows the loss on the test set, and bottom row shows the norm of the gradient on the training set.

# E  IMAGE CLASSIFICATION ON CONVOLUTIONAL NEURAL NETS

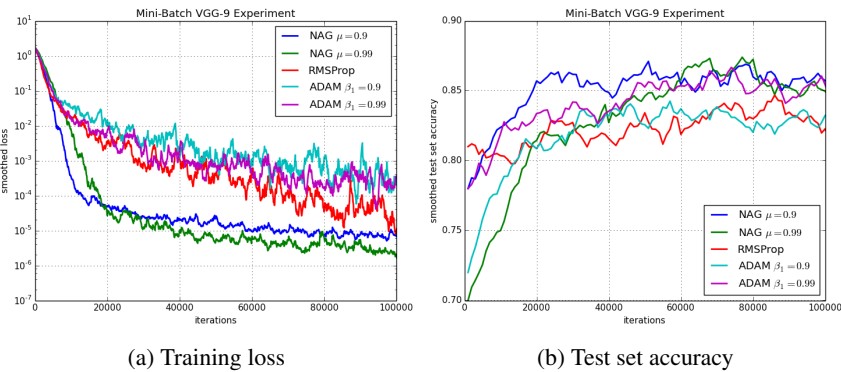

(a) Training loss        (b) Test set accuracy

Figure 12: Mini-batch image classification experiments with CIFAR-10 using VGG-9

To test whether these results might qualitatively hold for other datasets and models, we train an image classifier on CIFAR-10 (containing 10 classes) using VGG-like convolutional neural networks (Simonyan & Zisserman, 2014). In particular, we train VGG-9 on CIFAR-10, which contains 7 convolutional layers and 2 fully connected layers, a total of 9 layers. The convolutional layers contain 64, 64, 128, 128, 256, 256, 256 filters each of size $3 \times 3$, respectively. We use batch normalization (Ioffe & Szegedy, 2015) and ReLU activations after each convolutional layer, and the first fully connected layer. Table 1 contains more details of the VGG-9 architecture. We use minibatches of size 100, and weight decay of $10^{-5}$. We use fixed step sizes, and all hyperparameters were tuned as indicated in Section B.

We present results in Figure 12. As before, we see that this task is another example where tuning the momentum parameter ($\beta_1$) of ADAM helps. While attaining approximately the same loss value, ADAM with $\beta_1 = 0.99$ generalizes as good as NAG and better than when $\beta_1 = 0.9$. Thus tuning $\beta_1$ of ADAM helped in closing the generalization gap with NAG.

Table 1: VGG-9 on CIFAR-10.

| layer type | kernel size | input size | output size |
|---|---|---|---|
| Conv_1 | $3 \times 3$ | $3 \ \times 32 \times 32$ | $64 \ \times 32 \times 32$ |
| Conv_2 | $3 \times 3$ | $64 \ \times 32 \times 32$ | $64 \ \times 32 \times 32$ |
| Max Pooling | $2 \times 2$ | $64 \ \times 32 \times 32$ | $64 \ \times 16 \times 16$ |
| Conv_3 | $3 \times 3$ | $64 \ \times 16 \times 16$ | $128 \times 16 \times 16$ |
| Conv_4 | $3 \times 3$ | $128 \times 16 \times 16$ | $128 \times 16 \times 16$ |
| Max Pooling | $2 \times 2$ | $128 \times 16 \times 16$ | $128 \times 8 \ \times 8$ |
| Conv_5 | $3 \times 3$ | $128 \times 8 \ \times 8$ | $256 \times 8 \ \times 8$ |
| Conv_6 | $3 \times 3$ | $256 \times 8 \ \times 8$ | $256 \times 8 \ \times 8$ |
| Conv_7 | $3 \times 3$ | $256 \times 8 \ \times 8$ | $256 \times 8 \ \times 8$ |
| Max Pooling | $2 \times 2$ | $256 \times 8 \ \times 8$ | $256 \times 4 \ \times 4$ |
| Linear | $1 \times 1$ | $1 \times 4096$ | $1 \times 256$ |
| Linear | $1 \times 1$ | $1 \times 256$ | $1 \times 10$ |

