# OpenReview forum: "Convergence Guarantees for RMSProp and ADAM in Non-Convex Optimization and an Empirical Comparison to Nesterov Acceleration"
_ICLR.cc/2019/Conference_

### Official Review · AnonReviewer1 · 2018-10-26
**Missing a very relevant reference that discusses the exact same issue**

**Rating:** 5
**Confidence:** 4

**Review:**

*Summary:
This paper analyzes the convergence of ADAM and RMSProp to stationary points
in the non convex setting.
In the second part the authors experimantally compare the performance of these methods to Nesterov's Accelerated method.



*Comments:

-The paper does not tell a coherent story and the two parts of the paper are somewhat unrelated.

-The authors claim that they are the first to analyze adaptive methods in the non-convex setting, yet this was recently done in
[Xiaoyu Li, Francesco Orabona; On the Convergence of Stochastic Gradient Descent with Adaptive Stepsizes]
The authors should cite this paper and compare their results to it.

-The above paper of [Li and Orabona] demonstrates a nice benefit of AdaGrad in the non-convex setting. Concretely they show that in the noisless setting adaptive methods give a faster rate of $O(1/T)$ compared to the standard rate of $O(1/\sqrt{T})$ of SGD.

Unfortunately, the results of the current paper do not illustrate the benefit of adaptive methods over SGD, since the authors provide similar rates to SGD or even worse rates in some situations.
I think that in light of [Li and Orabona] one should expect a $O(1/T)$ rate also for ADAM and RMSProp.


-The experimental part is not so related to the first part. And the experimental phenomena is only demonstrated for the MNIST dataset, which is not satisfying.


*Summary:
The main contribution of this paper is to provide rates for approaching stationary points.
This is done for ADAM and RMSProp, two adaptive training methods.
The authors do not mention a very relevant reference, [Li and Orabona].
Also, the authors do not show if ADAM and RMSProp have any benefit compared to SGD in the non-convex setting, which is a bit disappointing. Especially since [Li and Orabona] do demonstrate the benefit of AdaGrad in their paper.

---

> ### Author Response · Authors · 2018-11-19
> **Added CIFAR-10 experiments and thanks for telling us about the Li-Orabona paper (and our comparisons to them)**
>
> Thanks a lot for your review. Taking your suggestion we have added in Appendix E (page 29) a mini-batch image classification experiment on CIFAR-10 (using VGG-9) and have shown that the basic conclusion we had on MNIST continues to hold: that ADAM has lower test-errors as beta-1 is properly tuned (and in our experiments, as it gets close to 1) and at that point it is among the best performers (in terms of test-loss) when compared to RMSProp and NAG.
>
> Also thanks a lot for bringing this paper ( https://arxiv.org/pdf/1805.08114.pdf) to our attention! We didnt know of this and its indeed a very beautiful paper on a related topic which seems to have appeared a couple of months before our work was completed.  We agree that they get the claim to be the first analysis for adaptive gradient methods on non-convex functions and we have now put in a citation for their paper. (Also on the point of similarities, as in Li-Orabona proofs we too do not need projection steps to get convergence.) But we would like to explain that the relative advantages of our results are quite significant.
>
> The algorithms they are analyzing are *not* the realistic RMSProp or ADAM which is what we consider. Unlike Li-Orabona, we demonstrate extensive experimental evidence in our paper to motivate the superiority on neural nets of the particular form of adaptivity that we write proofs about  - which infact is much more involved than the modification of AdaGrad that they consider.
>
> Most importantly Li-Orabona's analysis is not for "``fully" adaptive algorithms for a very specific reason as we explain in the first point below in the following list of 3 specific reasons as to why we are doing better than them.
>
> 1.
> Their non-asymptotic non-convex proof in Theorem 4 (page 7) can be seen as the closest analogue to our stochastic RMSProp proof. But they use a somewhat artificial form of AdaGrad as specified in their equation 3 (top of page 4).
> This particular choice of step-length (used in all but one of their proofs) essentially means that their adaptivity is uniform across all coordinates of the gradient! Hence they are removing a very important feature of adaptive algorithms that these algorithms scale different coordinates of the gradient by varying amounts! (empirically this feature is known to be extremely critical!)
>
> In contrast in our analysis we allow the pre-conditioner to act on the entire gradient vector and thus each coordinate of the gradient gets its own non-trivial scaling.
>
> Also Li-Orabona prevent the t-th preconditioner from depending on the t-th gradient.  From experiments on RMSProp we know that this modification can significantly hurt performance. We do not do such modifications!
>
>
>
> 2.
> Also their Theorem 4 (page 3) uses an epsilon > 0 setting (their epsilon is defined in their step-size definitions of equations 3 and 4 at the top of page 4). Now not only this means that theirs is not the "true" AdaGrad but to the best of our understanding , this also means that their convergence rate in Theorem 4 is *slower* than the standard rates that we can get for stochastic RMSProp (wth some assumptions on the training set) - which is a more complex algorithm!
>
>
> 3.
> You said "Concretely they show that in the noiseless setting adaptive methods give a faster rate of $O(1/T)$ compared to the standard rate of $O(1/\sqrt{T})$ of SGD" But apologies that we cannot locate any specific theorem in Li-Orabona which shows such a thing.
>
> The closest they come to such a result is their statement at the top of their page 6 where they say that under the assumption of their parameter sigma = 0 their convex stochastic adaptive proof is O(1/T) convergence as opposed to convex SGD's O(1/sqrt{T}) Kindly let us know if this is the statement in their paper that you had in mind.
>
> But this argument on the top of their page 6 is not convincing to us because when sigma =0 their hypothesis (H4') in Page 3 becomes undefined! And Assumption H4' is necessary for their Theorem 1 (bottom of Page 4). And maybe more importantly for *any* sigma > 0, however small, their upperbound in Theorem 1 will eventually be dominated by the second term in the max which is scaling upwards with T as T^{1/2 + epsilon}. Hence to the best of our understanding,  their convergence rates are guaranteed to be *slower* than O(1/sqrt{T}).
>
> We hope we have convinced you as to why our results are significantly better than those in the Li-Orabona paper.

---

> > ### Comment · AnonReviewer1 · 2018-12-03
> > **Comment**
> >
> > I thank the authors for their response.
> > Nevertheless, in face of the [Li and Orabona], I think that their contribution is incremental.
> >
> > - Indeed, only  when \sigma ->0, then [Li and Orabona] enable fast rate of 1/T.
> >   This is relevant to stochastic settings where we use large batch sizes which decrease the variance. I don't see why this contradicts hypothesis (H4') in Page 3 (Just take S=0).
> >
> > - Again, the paper does not tell a coherent story and the two parts of the paper are somewhat unrelated.
> >
> > I therefore keep my score.

---

### Official Review · AnonReviewer2 · 2018-10-30
**There may exist an error on the proof of Stochastic RMSProp.**

**Rating:** 4
**Confidence:** 5

**Review:**

There may exist an error on the proof of Theorem 3.1 in appendix. For the first equation in page 13, the authors want to estimate lower-bound of the term $E<\nabla{f}(xt),V_t^{-0/5}*gt>$. The second inequality $>$ may be wrong. Please check it carefully.  (Hints: both the index sets { i | \nabla{f}(xt))_{i}*gt_{i} <0 } and { i | \nabla{f}(xt))_{i}*gt_{i} >0 } depend on the random variable $gt$. Hence, the expectation and summation cannot be exchanged in the second inequality.)

---

> ### Author Response · Authors · 2018-11-19
> **Clarifications about our Theorem 3.1**
>
> Thanks a lot for your careful reading of our proofs and for bringing to our attention this one ambiguous step we had in the proof of Theorem 3.1 Indeed we messed up at that step. We would like to express our sincerest gratitude to you for bringing this to our attention. Kindly see the edited Theorem 3.1 in the revised file that we have uploaded - whereby we have now imposed a certain technical condition about the gradients of the functions contributing to the empirical loss. We note that this condition in no way makes the proof trivial.
>
> Now we have a somewhat detailed new Lemma A.1 starting in the middle of page 13. This lemma is invoked inside the proof (at the location where you raised your doubts) and the rest of the proof and the theorem's final guarantees remain unchanged.
>
> To the best of our knowledge is the the first proof giving any non-trivial sufficient conditions for convergence to criticality for stochastic RMSProp.
>
> Kindly also see the new VGG-9 with CIFAR-10 experiments in Appendix E (page 29) that we have now added based on suggestions from the Reviewer 1. Hopefully we have now convinced you of the correctness of our results and why they are very interesting.

---

### Official Review · AnonReviewer3 · 2018-11-02
**Nice idea , not so good presentation**

**Rating:** 5
**Confidence:** 3

**Review:**

Summary:
This paper present a convergence analysis of the popular methods RMSProp and ADAM in the case of smooth non-convex functions. In particular it was shown that the above adaptive gradient algorithms are guaranteed to reach critical points for smooth non-convex objectives and bounds on the running time are provided. An empirical investigation is also presented with main focus on the comparison of the adaptive gradient methods and the Nesterov accelerated gradient algorithm (NAG).

Comments:
Although the results are promising, I found the reading (mainly because of the not defined notation) of this paper really hard.
In terms of presentation, the motivation in introduction is fine, but the following section named "Notations and Pseudocodes" is confusing and has many undefined notations which makes the paper very hard to read. It gives the impression that the section was added the last minute. For example what is fundtion "g" in the definition 1? What is support(v) and the diag(v) in the definition 2. the diag(v) is more obvious to me but then why at page 18 the diag(v)at the top of the page is bold (are these two things different)?
In the presentation of RMSProp what the $g_t^2$ means? Please have a look to last year's ICLR paper [Reddi, Sashank J., Satyen Kale, and Sanjiv Kumar. "On the convergence of adam and beyond." (2018).] for a more appropriate introduction of the notation.

In the introduction the authors refer to NAG as a stochastic variant of the Nesterov's acceleration and they informally present the algorithm in the end of the first paragraph. There the update rule includes stochastic gradients \nable f_i(.) while in the formal presentation in the update rule there is \nabla f(x) which is the full gradient of the objective function of the original problem. I expect this difference is somehow justified from the mentioning in the algorithm of the possibly noisy oracle but this is never mention in the main text.

If the above statements in terms of presentation, are ignored the convergence results and numerical experiments are interesting.
However, the numerical evaluation does not correspond to the theoretical results. It is a comparison of NAG ,ADAM and RMSPROP with interesting conclusions  that can be beneficial for practitioners that they use these methods.

Some missing references:
On Adam methods:
1) Chen, Xiangyi, et al. "On the convergence of a class of adam-type algorithms for non-convex optimization." arXiv preprint arXiv:1808.02941 (2018).
2) Zhou, Dongruo, et al. "On the convergence of adaptive gradient methods for nonconvex optimization." arXiv preprint arXiv:1808.05671 (2018).
On momentum (heavy ball) methods:
3) Loizou, Nicolas, and Peter Richtárik. "Momentum and stochastic momentum for stochastic gradient, Newton, proximal point and subspace descent methods." arXiv preprint arXiv:1712.09677 (2017).

---

> ### Author Response · Authors · 2018-11-19
> **Clarification about the notational issues raised.**
>
> Thanks for your detailed review.
> In this note let me try to quickly clarify the notation issues that you have raised.
>
> 1.
> In Definition 1(page 3) "g" is a typo! Apologies for this! We have now fixed this typo.
> It should be a "f" in terms of which the L-smoothness' defining inequality has been given there.
>
> 2.
> In Definition 2(page 3) "Support(v)" denotes the set of indices of the d-dimensional vector v where the corresponding coordinate of v is non-zero. (...indeed here we are defining the matrix V^(-1/2) using a Penrose inverse and this is somewhat different from say the notation in the ICLR 2018 paper that you mention (https://openreview.net/forum?id=ryQu7f-RZ)  because we wanted to have an unifying notation between not just RMSProp and ADAM but also for our Theorem 3.3 where there is no \xi parameter...)
>
> 3.
> There is no conceptual difference between the "diag" in Definition 2 (page 3) and the bold-faced diag that one sees in the paragraph headings on page 18 - this bold-facing in the later instance comes from LaTeX's default settings about how it prints headings in the  \paragraph{} environment.
>
> 4.
> The \nabla f(x) which occurs in our Algorithm 1 and 2 pseudocodes (on page 3) is not to be understood as necessarily being a full gradient. It is a notational stand-in for whatever the first-order oracle returns and this oracle could be noisy as we specify in the "Input" lines of these two algorithms.
>
> In the first paragraph (page 1) when we give an informal description of NAG our use the notation \nabla \tilde{f}_{i_t}  (distinct from the pseudocodes) is adapted to the specific model of stochasticity that we use in that paragraph which is specified as choosing \tilde{f}_{i_t} randomly from the set of functions {f_1,..,f_k} over which the empirical risk is taking the average.
>
> 5.
> Both in RMSProp as well as ADAM our notation "g_t^2" refers to a vector whose i-th coordinate is the square of the i-th coordinate of the vector g_t i.e if g_t \in R^d then for all i in {1,..,d}, ((g_t)^2)_i = ((g_t)_i)^2
>
> Thanks for pointing out the references.  We have now incorporated these references into a revised upload. Also we noticed that both these references (arXiv:1808.02941 and arXiv:1808.05671) came out after the first version of our work became public and infact both of them cite us.
>
> Kindly also see the new VGG-9 with CIFAR-10 experiments in Appendix E (page 29) that we have now added based on suggestions from the Reviewer 1. Hopefully we have now convinced you of the correctness of our results and why they are very interesting.

---

### Public Comment · ~Jeremy_Bernstein1 · 2018-11-12
**Clarifying the relevance of some prior work**

Hi there, I want to clarify the relevance of some prior work.

The authors say:

"To the best of our knowledge, this work gives the first convergence guarantees for adaptive gradient algorithms in the context of non-convex optimization. We show run-time bounds for (stochastic and deterministic) RMSProp and deterministic ADAM to reach approximate criticality on smooth non-convex functions."

But (Bernstein et al. 2018 a) have proved non-convex, stochastic convergence bounds for signSGD which is equivalent to Adam with all momentum switched off. Therefore the above statement from the authors seems like it may deserve at least some qualification.

What's more since signSGD is a special case of Adam, it seems fair to ask the authors to analyse and discuss how their work relates to the signSGD work.

Admittedly attacking Adam is harder than signSGD. But it seems like a good strategy for attacking general stochastic Adam would be to extend the signSGD result. There is a refined analysis of signSGD in this paper: (Bernstein et al. 2018 b) which should be of interest. It seems fair not to expect the authors to know about this second work since it came out so recently.

(Bernstein et al. 2018 a) https://arxiv.org/abs/1802.04434
(Bernstein et al. 2018 b) https://arxiv.org/abs/1810.05291

---

> ### Author Response · Authors · 2018-11-19
> **Thanks for sharing your thoughts!**
>
> Thanks a lot for your detailed comments! We had seen your beautiful paper (https://arxiv.org/abs/1802.04434) and had already referred to that in our introduction. We critically use your paper in motivating our setup. We hadnt seen your second paper when we completed our work.
>
> As Reviewer1 pointed out to us it seems that there has been this paper (https://arxiv.org/abs/1805.08114) a couple of months before our work was done and hence they rightfully get the credit for being the first proofs of convergence of adapative gradient methods in non-convex settings. But as we have explained in our response to Reviewer 1 below, that there are a number of reasons why we feel ours is a more natural setup and our results are much more substantial than the Li-Orabona paper.
>
> Your paper was definitely a few months before even the Li-Orabona paper but I think sign-based updates which are doing gradient compression should be thought of as a different category than gradient based updates. To our mind, these are conceptually different things.
>
> Ofcourse there seems to be some underlying connection between the sign pattern of the gradients and convergence of stochastic adaptive methods as evidenced by the kind of assumptions we need to prove that stochastic RMSProp gets to criticality at standard speeds. To the best of our knowledge ours is the first theorem of its kind which gives some non-trivial sufficient conditions for stochastic RMSProp to get to criticality of non-convex functions. This is definitely a direction to look further into and understand this better. We look forward to more exchange of ideas between the two approaches!

---

### Author Response · Authors · 2018-11-19
**A summary of the edits in the revised version.**

We would like to express our sincere gratitude for the reviewers whose valuable feedback has helped improve the paper as reflected in the revised update. We have responded to their specific queries in the individual responses below.

In the revised version of the paper that we have uploaded we have made 2 main edits as follows,
(a) Now there is a new Appendix E showing that VGG-9 running on CIFAR-10 continues to demonstrate the interesting beta_1 sensitivity of ADAM that we had previously explored for autoencoders running on MNIST
(b) The statement of the Theorem 3.1 has been refined and its proof in Appendix A.1 has been appropriately updated. Now there is a somewhat elaborate Lemma A.1 which helps clarify some issues.

---

### Meta-Review · Area_Chair1 · 2018-12-17

**Confidence:** 4
**Recommendation:** Reject

**Metareview:**

The reviewers and ACs acknowledge that the paper has a solid theoretical contribution  because it give a convergence (to critical points) of the ADAM and RMSprop algorithms, and also shows that NAG can be tuned to match or outperform SGD in test errors. However, reviewers and the AC also note that potential improvements for the paper a) the exposition/notations can be improved; b) better comparison to the prior work could be made; c) the theoretical and empirical parts of the paper are somewhat disconnected; d) the proof has an error (that is fixed by the authors with additional assumptions.) Therefore, the paper is not quite ready for publications right now but the AC encourages the authors to submit revisions to other top ML venues.